# Incorporating Arbitrary Matrix Group Equivariance into KANs

**Lexiang Hu** [1]   **Yisen Wang** [1,2]   **Zhouchen Lin** [1,2,3]

## Abstract

Kolmogorov-Arnold Networks (KANs) have seen great success in scientific domains thanks to spline activation functions, becoming an alternative to Multi-Layer Perceptrons (MLPs). However, spline functions may not respect symmetry in tasks, which is crucial prior knowledge in machine learning. In this paper, we propose Equivariant Kolmogorov-Arnold Networks (EKAN), a method for incorporating arbitrary matrix group equivariance into KANs, aiming to broaden their applicability to more fields. We first construct gated spline basis functions, which form the EKAN layer together with equivariant linear weights, and then define a lift layer to align the input space of EKAN with the feature space of the dataset, thereby building the entire EKAN architecture. Compared with baseline models, EKAN achieves higher accuracy with smaller datasets or fewer parameters on symmetry-related tasks, such as particle scattering and the three-body problem, often reducing test MSE by several orders of magnitude. Even in non-symbolic formula scenarios, such as top quark tagging with three jet constituents, EKAN achieves comparable results with state-of-the-art equivariant architectures using fewer than $40\%$ of the parameters, while KANs do not outperform MLPs as expected. Code and data are available at https://github.com/hulx2002/EKAN.

## 1. Introduction

Kolmogorov-Arnold Networks (KANs) (Liu et al., 2024b;a) are a novel type of neural network inspired by the Kolmogorov-Arnold representation theorem (Tikhomirov,

[1]State Key Lab of General AI, School of Intelligence Science and Technology, Peking University [2]Institute for Artificial Intelligence, Peking University [3]Pazhou Laboratory (Huangpu), Guangzhou, Guangdong, China. Correspondence to: Zhouchen Lin <zlin@pku.edu.cn>.

*Proceedings of the $42^{nd}$ International Conference on Machine Learning*, Vancouver, Canada. PMLR 267, 2025. Copyright 2025 by the author(s).

1991; Braun & Griebel, 2009), which offers an alternative to Multi-Layer Perceptrons (MLPs) (Haykin, 1998; Cybenko, 1989; Hornik et al., 1989). Unlike MLPs, which utilize fixed activation functions on nodes, KANs employ learnable activation functions on edges, replacing the linear weight parameters entirely with univariate functions parameterized as splines (De Boor, 1978). On the other hand, each layer of KANs can be viewed as spline basis functions followed by a linear layer (Dhiman, 2024). This architecture allows KANs to achieve better accuracy in symbolic formula representation tasks compared with MLPs, particularly in function fitting and scientific applications. Subsequent works based on KANs have demonstrated superior performance in other areas, such as sequential data (Vaca-Rubio et al., 2024; Genet & Inzirillo, 2024b;a; Xu et al., 2024), graph data (Bresson et al., 2024; De Carlo et al., 2024; Kiamari et al., 2024; Zhang & Zhang, 2024), image data (Cheon, 2024b;a; Azam & Akhtar, 2024; Li et al., 2024a; Seydi, 2024; Bodner et al., 2024), and so on.

However, KANs themselves perform poorly on non-symbolic formula representation tasks (Yu et al., 2024). One reason for this is that splines struggle to respect data type and symmetry, both of which play important roles in machine learning. Many recent works utilize symmetry in data to design network architectures, achieving better efficiency and generalization on specific tasks. For example, Convolutional Neural Networks (CNNs) (LeCun et al., 1989) and Group equivariant Convolutional Neural Networks (GC-NNs) (Cohen & Welling, 2016) leverage translational and rotational symmetries in image data, while DeepSets (Zaheer et al., 2017) and equivariant graph networks (Maron et al., 2019; Keriven & Peyré, 2019; Satorras et al., 2021) exploit the permutation symmetry in set and graph data. Equivariant Multi-Layer Perceptrons (EMLP) (Finzi et al., 2021) propose a general method that allows MLPs to be equivariant with respect to arbitrary matrix groups for specific data types, thereby unifying the aforementioned specialized network architectures.

Inspired by these equivariant architectures, we propose Equivariant Kolmogorov-Arnold Networks (EKAN), which embed matrix group equivariance into KANs. By specifying the data type and symmetry, EKAN can serve as a general framework for applying KANs to various areas. In Section 2, we introduce the preliminary knowledge of group theory. In

Section 3, we summarize related works. In Section 4, we construct a layer of EKAN. We add gate scalars to the input and output space of each layer, and define gated spline basis functions between the input and post-activation space. To ensure equivariance when linearly combining gated basis functions, we construct the equivariant constraint and solve for the equivariant linear weights. In Section 5, we build the entire EKAN architecture. We insert a lift layer before the first layer and discard the gate scalars from the output of the final layer, so that the input and output space of EKAN can be consistent with the original dataset. In Section 6, we evaluate EKAN on tasks with known symmetries. We show that EKAN can achieve higher accuracy than baseline models with smaller datasets or fewer parameters. In Section 7, we conclude this work. In Figure 1, we compare the architectures of KANs and EKAN.

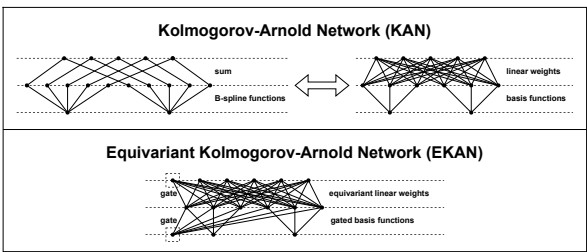

*Figure 1.* Comparison of the architectures of Kolmogorov-Arnold Networks (KANs) and Equivariant Kolmogorov-Arnold Networks (EKAN).

In summary, our contributions are as follows: (1) We propose EKAN, an architecture that makes KANs equivariant to matrix groups. To our knowledge, EKAN is the first attempt to combine equivariance with KANs, and we expect that it can serve as a general framework to broaden the applicability of KANs to more areas. (2) We specify the space structures of the EKAN Layer and define gated spline basis functions. We theoretically prove that gated basis functions can ensure equivariance between the gated input space and the post-activation space. Then, we insert a lift layer to preprocess the raw input feature, which aligns the input space of EKAN with the feature space of the dataset. (3) Experiments on tasks with matrix group equivariance, such as particle scattering and the three-body problem, demonstrate that EKAN often significantly outperforms baseline models, even with smaller datasets or fewer parameters. In the task of non-symbolic formula representation, where KANs are not proficient, such as top quark tagging with three jet constituents, EKAN can still achieve comparable results with state-of-the-art equivariant architectures while using fewer than 40% of the parameters.

## 2. Background

Before presenting related works and our method, we first introduce some preliminary knowledge of group theory.

**Groups and generators.** The matrix group $\widetilde{G}$ is a subgroup of the general linear group $\mathrm{GL}(n)$, which consists of $n \times n$ invertible matrices. Each group element $g \in \widetilde{G}$ can be decomposed into a continuous and a finite component $g = g_1 g_2$. We can obtain the continuous component $g_1$ from a Lie algebra element $A \in \mathfrak{g}$ through the exponential map $\exp : \mathfrak{g} \to \widetilde{G}$, i.e., $g_1 = \exp(A) = \sum_{k=0}^{\infty} \frac{A^k}{k!}$. Representing the space where the Lie algebra resides as a basis $\{A_i\}_{i=1}^{D}$, we have $g_1 = \exp\left(\sum_{i=1}^{D} \alpha_i A_i\right)$. On the other hand, the finite component $g_2$ can be generated by a set of group elements $\{h_i\}_{i=1}^{M}$ and their inverses $h_{-k} = h_k^{-1}$, formally speaking $g_2 = \prod_{i=1}^{N} h_{k_i}$. Overall, we can express the matrix group element as:

$$g = \exp\left(\sum_{i=1}^{D} \alpha_i A_i\right) \prod_{i=1}^{N} h_{k_i}, \tag{1}$$

where $\{A_i\}_{i=1}^{D}$ are called infinitesimal generators and $\{h_i\}_{i=1}^{M}$ are called discrete generators. We introduce common matrix groups and their generators in Appendix A.

**Group representations.** The group representation $\rho_V : \widetilde{G} \to \mathrm{GL}(m)$ maps group elements to $m \times m$ invertible matrices, which describes how group elements act on the vector space $V = \mathbb{R}^m$ through linear transformations. For $g_1, g_2 \in \widetilde{G}$ it satisfies $\rho_V(g_1 g_2) = \rho_V(g_1)\rho_V(g_2)$. Similarly, the Lie algebra representation is defined as $\mathrm{d}\rho_V : \mathfrak{g} \to \mathfrak{gl}(m)$, and for $A_1, A_2 \in \mathfrak{g}$, we have $\mathrm{d}\rho_V(A_1 + A_2) = \mathrm{d}\rho_V(A_1) + \mathrm{d}\rho_V(A_2)$. We can relate the Lie group representation to the Lie algebra representation through the exponential map. Specifically, for $A \in \mathfrak{g}$, $\rho_V(\exp(A)) = \exp(\mathrm{d}\rho_V(A))$ holds. Then, combining with Equation (1), the matrix group representation can be written as:

$$\rho_V(g) = \exp\left(\sum_{i=1}^{D} \alpha_i \mathrm{d}\rho_V(A_i)\right) \prod_{i=1}^{N} \rho_V(h_{k_i}). \tag{2}$$

We can construct the complex vector space from the base vector space using the dual ($*$), direct sum ($\oplus$), and tensor product ($\otimes$) operations. To give a concrete example, let $V_1$ and $V_2$ be base vector spaces. The multi-channel vector space, matrix space and parameter space of the linear mapping $V_1 \to V_2$ can be represented as $V_1 \oplus V_2$, $V_1 \otimes V_2$, and $V_2 \otimes V_1^*$, respectively. In general, given a matrix group $\widetilde{G}$, we can normalize a vector space $U$ into a polynomial-like form with respect to the base vector space $V$ of $\widetilde{G}$ (the space where the group representation is the identity mapping $\rho_V(g) = g$; intuitively, the transformation matrix is

the matrix group element itself):

$$U = \bigoplus_{a=1}^{A} T(p_a, q_a) = \bigoplus_{a=1}^{A} V^{p_a} \otimes (V^*)^{q_a}, \quad (3)$$

where $V^{p_a} = \underbrace{V \otimes V \otimes \cdots \otimes V}_{p_a}$ and $(V^*)^{q_a} = \underbrace{V^* \otimes V^* \otimes \cdots \otimes V^*}_{q_a}$. Its group representation and Lie algebra representation can be generated by the following rules:

$$\begin{cases} \rho_{V^*}(g) = \rho_V(g^{-1})^\top, \\ \rho_{V_1 \oplus V_2}(g) = \rho_{V_1}(g) \oplus \rho_{V_2}(g), \\ \rho_{V_1 \otimes V_2}(g) = \rho_{V_1}(g) \otimes \rho_{V_2}(g), \\ \mathrm{d}\rho_{V^*}(A) = -\mathrm{d}\rho_V(A)^\top, \\ \mathrm{d}\rho_{V_1 \oplus V_2}(A) = \mathrm{d}\rho_{V_1}(A) \oplus \mathrm{d}\rho_{V_2}(A), \\ \mathrm{d}\rho_{V_1 \otimes V_2}(A) = \mathrm{d}\rho_{V_1}(A) \boxplus \mathrm{d}\rho_{V_2}(A), \end{cases} \quad (4)$$

where $\oplus$ is the direct sum, $\otimes$ is the Kronecker product, and $\boxplus$ is the Kronecker sum. We provide concrete examples of the space structure in Appendix B to help readers understand it intuitively.

**Equivariance and invariance.** The symmetry can be divided into equivariance and invariance, meaning that when a transformation is applied to the input space, the output space either transforms in the same way or remains unchanged. Formally, given a group $\widetilde{G}$, a function $f : U_i \to U_o$ is equivariant if:

$$\forall g \in \widetilde{G}, v_i \in U_i : \quad \rho_o(g) f(v_i) = f(\rho_i(g) v_i), \quad (5)$$

where $\rho_i$ and $\rho_o$ are group representations of $U_i$ and $U_o$, respectively. Specifically, when $\rho_o(g) = I$, the function $f$ is invariant.

## 3. Related Works

**Equivariant networks.** Equivariant networks have gained significant attention in recent years due to their ability to respect and leverage symmetries in data. GCNNs (Cohen & Welling, 2016) embed discrete group equivariance into traditional CNNs through group convolutions. Steerable CNNs (Cohen & Welling, 2017) introduce steerable filters, which provide a more flexible and efficient way to achieve equivariance compared with GCNNs. Subsequently, SFCNNs (Weiler et al., 2018b) and E(2)-equivariant steerable CNNs (Weiler & Cesa, 2019) extend GCNNs and steerable CNNs to continuous group equivariance, while 3D Steerable CNNs (Weiler et al., 2018a) extend these models to 3D volumetric data. On the other hand, some works use partial differential operators (PDOs) to construct equivariant networks (Shen et al., 2020; 2021; 2022; He et al., 2022; Li et al., 2024b; 2025). Furthermore, equivariant self-supervised learning

(Wang et al., 2020; Dangovski et al., 2022; Lee et al., 2022; Garrido et al., 2023; Gupta et al., 2024) has also achieved outstanding results. Based on these theoretical frameworks, equivariant networks are widely applied in various fields, such as mathematics (Zhao et al., 2023), physics (Wang et al., 2021; Hu et al., 2025), biochemistry (Bekkers et al., 2018; Winkels & Cohen, 2019; Graham et al., 2020), and others. EMLP (Finzi et al., 2021) embeds matrix group equivariance into MLPs layerwise, which we discuss in detail in Appendix C.

**Kolmogorov-Arnold Networks (KANs).** KANs (Liu et al., 2024b;a) place learnable activation functions on the edges and then sum them to obtain the output nodes, replacing the fixed activation functions applied to the output nodes of linear layers in MLPs. Formally, the $l$-th KAN layer can be expressed as:

$$x_{l+1,j} = \sum_{i=1}^{n_l} \phi_{l,j,i}(x_{l,i}), \quad j = 1, \ldots, n_{l+1}, \quad (6)$$

where $n_l$ is the number of nodes in the $l$-th layer, $x_{l,i}$ is the value of the $i$-th node in the $l$-th layer, and $\phi_{l,j,i}$ is the activation function that connects $x_{l,i}$ to $x_{l+1,j}$. In practice, $\phi_{l,j,i}$ consists of a spline function and a silu function. The spline basis functions are determined by grids, which are updated based on the input samples. Then we can write the post-activation of $\phi_{l,j,i}$ as $\phi_{l,j,i}(x_{l,i}) = \sum_{b=0}^{G+k-1} w_{l,j,i,b} B_{l,i,b}(x_{l,i}) + w_{l,j,i,G+k} \mathrm{silu}(x_{l,i})$, where $G$, $k$, and $B_{l,i,b}$ represent the number of grid intervals, the order, and the $b$-th basis function of splines at node $x_{l,j}$, respectively. Therefore, the KAN layer can be viewed as spline basis functions $B_{l,i,b}$ and a silu function, followed by a linear layer with $w_{l,j,i,b}$ as parameters (Dhiman, 2024):

$$x_{l+1,j} = \sum_{i=1}^{n_l} \left[ \sum_{b=0}^{G+k-1} w_{l,j,i,b} B_{l,i,b}(x_{l,i}) \right.$$
$$\left. + w_{l,j,i,G+k} \mathrm{silu}(x_{l,i}) \right], \quad j = 1, \ldots, n_{l+1}. \quad (7)$$

## 4. EKAN Layer

In this section, we construct the EKAN layer, which is equivariant with respect to the matrix group $\widetilde{G}$. First, we define the space structures and explain their relationships. Then, we introduce gated basis functions and equivariant linear weights, which together form a layer of EKAN. We summarize the space structures and network architecture of the EKAN layer in Figure 2.

### 4.1. Space Structures

A key aspect of equivariant networks is how group elements act on the feature space. Therefore, unlike conventional

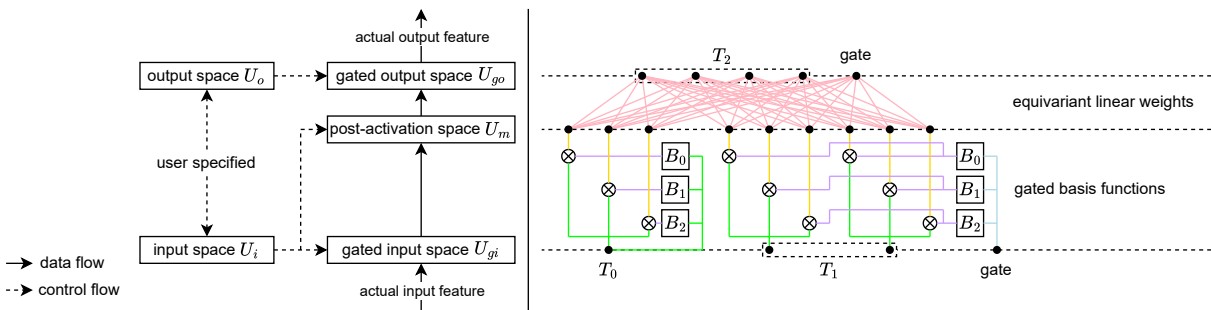

*Figure 2.* (Left) The space structures of the EKAN layer and their relationships. (Right) The architecture of the EKAN layer, which consists of gated basis functions and equivariant linear weights.

networks, which only focus on the dimensions of the feature space, equivariant networks need to further clarify the structure of the feature space. For example, for the group SO(2), two feature spaces $U_1 = V \oplus V = \mathbb{R}^2 \oplus \mathbb{R}^2$ and $U_2 = V \otimes V = \mathbb{R}^2 \otimes \mathbb{R}^2$ have different group representations $\rho_{U_1}$ and $\rho_{U_2}$, but conventional networks treat them as the same space $U = \mathbb{R}^4$.

We specify the input space and the output space of EKAN layer as $U_i$ and $U_o$, respectively. Their structures can be normalized into the form of Equation (3). In particular, for ease of later discussion, we extract the scalar space terms $T_0 = T(0,0)$ and rewrite them as:

$$\begin{cases} U_i = c_i T_0 \oplus \left[ \bigoplus_{a=1}^{A_i} T(p_{i,a}, q_{i,a}) \right], \\ U_o = c_o T_0 \oplus \left[ \bigoplus_{a=1}^{A_o} T(p_{o,a}, q_{o,a}) \right], \end{cases} \quad (8)$$

where $p_{i,a}, q_{i,a}, p_{o,a}, q_{o,a}, c_i, c_o, A_i, A_o \in \mathbb{N}$, $p_{i,a} + q_{i,a} > 0$, $p_{o,a} + q_{o,a} > 0$, and $cT_0 = \underbrace{T_0 \oplus T_0 \oplus \cdots \oplus T_0}_{c}$.

We have to emphasize that the actual input/output feature does not lie within $U_i/U_o$. To align with gated basis functions, we add a gate scalar $T_0$ to each non-scalar term $T(p_{i,a}, q_{i,a})/T(p_{o,a}, q_{o,a})$ in $U_i/U_o$ to obtain the actual input/output space. The mechanism behind this approach will be discussed in detail in Section 4.2. We denote this actual input/output space as the gated input/output space $U_{gi}/U_{go}$:

$$\begin{cases} U_{gi} = c_i T_0 \oplus \left[ \bigoplus_{a=1}^{A_i} T(p_{i,a}, q_{i,a}) \right] \oplus A_i T_0, \\ U_{go} = c_o T_0 \oplus \left[ \bigoplus_{a=1}^{A_o} T(p_{o,a}, q_{o,a}) \right] \oplus A_o T_0. \end{cases} \quad (9)$$

As shown in Equation (7), we split the KAN layer into basis functions and linear weights. From this perspective, we correspondingly construct gated basis functions and equivariant linear weights to form the EKAN layer. We refer to the space where the activation values reside after gated basis functions and before equivariant linear weights as the post-activation space $U_m$. The structure of $U_m$ depends on $U_i$, which we will elaborate on in Section 4.2.

We summarize the aforementioned space structures and their relationships in Figure 2 (Left). The user first specifies the input space $U_i$ and the output space $U_o$ for the EKAN layer. Then the gated input space $U_{gi}$ and the post-activation space $U_m$ are calculated based on $U_i$, and the gated output space $U_{go}$ is calculated based on $U_o$. The actual input feature in $U_{gi}$ passes through gated basis functions to obtain the activation value in $U_m$, which then passes through equivariant linear weights to obtain the actual output feature in $U_{go}$.

### 4.2. Gated Basis Functions

Since spline basis functions are nonlinear and have relatively complex iterative forms, directly solving the equivariance constraint in Equation (5) is quite challenging. Recently, gating mechanisms have been widely used in various areas, not only in language models to improve performance (Dauphin et al., 2017; Shazeer, 2020; Gu & Dao, 2023) but also in the design of equivariant networks (Weiler et al., 2018a; Finzi et al., 2021). Inspired by this, we propose gated basis functions to make the basis functions in KANs (spline basis functions and the silu function) equivariant. Suppose that the input feature $v_{gi} \in U_{gi}$ can be decomposed according to the space structure shown in Equation (9):

$$v_{gi} = \left( \bigoplus_{a=1}^{c_i} s_{i,a} \right) \oplus \left( \bigoplus_{a=1}^{A_i} v_{i,a} \right) \oplus \left( \bigoplus_{a=1}^{A_i} s'_{i,a} \right), \quad (10)$$

where $s_{i,a}, s'_{i,a} \in T_0$ and $v_{i,a} \in T(p_{i,a}, q_{i,a})$. For the non-scalar term $v_{i,a}$, we apply the basis functions to the corresponding gate scalar $s'_{i,a}$ and then multiply the result by $v_{i,a}$. For the scalar term $s_{i,a}$, we consider it as its own gate scalar, which is equivalent to applying basis functions element-wise to $s_{i,a}$. Formalizing the above content, the post-activation value $v_m \in U_m$ can be written as:

$$v_m = \bigoplus_{b=0}^{G+k} v_{m,b}, \quad (11)$$

where

$$v_{m,b} = \begin{cases} \left[\bigoplus_{a=1}^{c_i} s_{i,a} B_b(s_{i,a})\right] \oplus \left[\bigoplus_{a=1}^{A_i} v_{i,a} B_b(s'_{i,a})\right], \\ \qquad\qquad\qquad\qquad\qquad b < G+k, \\ \left[\bigoplus_{a=1}^{c_i} s_{i,a}\mathrm{silu}(s_{i,a})\right] \oplus \left[\bigoplus_{a=1}^{A_i} v_{i,a}\mathrm{silu}(s'_{i,a})\right], \\ \qquad\qquad\qquad\qquad\qquad b = G+k. \end{cases} \tag{12}$$

Note that $v_{m,b} \in c_i T_0 \oplus \left[\bigoplus_{a=1}^{A_i} T(p_{i,a}, q_{i,a})\right] = U_i$. Therefore, we obtain the structure of the post-activation space $U_m$:

$$U_m = (G+k+1)U_i. \tag{13}$$

The following theorem guarantees the equivariance between the gated input space and the post-activation space (see Appendix D for proof).

**Theorem 4.1.** *Given a matrix group $\widetilde{G}$, the gated input space $U_{gi}$ and the post-activation space $U_m$ can be expressed in the forms of Equations (9) and (13), respectively. The function $f : U_{gi} \to U_m$ is defined by Equations (10) to (12), that is, $v_m = f(v_{gi})$. Then, $f$ is equivariant:*

$$\forall g \in \widetilde{G}, v_{gi} \in U_{gi}: \quad \rho_m(g)f(v_{gi}) = f(\rho_{gi}(g)v_{gi}), \tag{14}$$

*where $\rho_{gi}$ and $\rho_m$ are group representations of $U_{gi}$ and $U_m$, respectively.*

### 4.3. Equivariant Linear Weights

The output feature $v_{go} \in U_{go}$ is obtained by a linear combination of the post-activation value $v_m \in U_m$. Let $U_i = \mathbb{R}^{d_i}$ and $U_{go} = \mathbb{R}^{d_{go}}$, then Equation (13) indicates that $U_m = \mathbb{R}^{(G+k+1)d_i}$. The linear weight matrix $W \in \mathbb{R}^{d_{go} \times (G+k+1)d_i}$ can be partitioned as $W = [W_0 \ W_1 \ \ldots \ W_{G+k}]$, where $W_b \in \mathbb{R}^{d_{go} \times d_i}$. Combining with Equation (11), we have:

$$v_{go} = Wv_m = \sum_{b=0}^{G+k} W_b v_{m,b}. \tag{15}$$

To ensure the equivariance between the post-activation space and the gated output space, we obtain:

$$\forall g \in \widetilde{G}, v_m \in U_m: \quad \rho_{go}(g)Wv_m = W\rho_m(g)v_m, \tag{16}$$

where $\rho_{go}$ is the group representation of $U_{go}$. Using the structure of $U_m$ shown in Equation (13) and applying the rules from Equation (4), we can derive that $\rho_m(g) = \bigoplus_{b=0}^{G+k} \rho_i(g)$, where $\rho_i$ is the group representation of $U_i$. Therefore, from Equation (11), we have $\rho_m(g)v_m = \bigoplus_{b=0}^{G+k} \rho_i(g)v_{m,b}$. Then Equation (16) can be written as:

$$\forall g \in \widetilde{G}, \{v_{m,b}\}_{b=0}^{G+k} \in U_i:$$
$$\sum_{b=0}^{G+k} \rho_{go}(g)W_b v_{m,b} = \sum_{b=0}^{G+k} W_b \rho_i(g)v_{m,b}. \tag{17}$$

The coefficients of each term in $\{v_{m,b}\}_{b=0}^{G+k}$ are equal:

$$\forall g \in \widetilde{G}, b \in \{0, 1, \ldots, G+k\}: \quad \rho_{go}(g)W_b = W_b\rho_i(g). \tag{18}$$

Flattening the linear weight blocks $\{W_b\}_{b=0}^{G+k}$ into vectors, we obtain:

$$\forall g \in \widetilde{G}, b \in \{0, 1, \ldots, G+k\}:$$
$$\rho_{go,i}(g)\mathrm{vec}(W_b) = \mathrm{vec}(W_b), \tag{19}$$

where $\rho_{go,i} = \rho_{go} \otimes \rho_i^*$ is the group representation of $U_{go} \otimes U_i^*$. We use the same method as EMLP (Finzi et al., 2021) to solve for the equivariant basis $Q$ and equivariant projector $P = QQ^\top$ of $\mathrm{vec}(W_b)$. Similar to the transition from Equation (27) to Equation (28) (in Appendix C), we decompose the group representation $\rho_{go,i}(g)$ into discrete and infinitesimal generators to obtain the following constraint:

$$\forall b \in \{0, 1, \ldots, G+k\}:$$
$$C\mathrm{vec}(W_b) = \begin{bmatrix} \mathrm{d}\rho_{go,i}(A_1) \\ \vdots \\ \mathrm{d}\rho_{go,i}(A_D) \\ \rho_{go,i}(h_1) - I \\ \vdots \\ \rho_{go,i}(h_M) - I \end{bmatrix} \mathrm{vec}(W_b) = 0. \tag{20}$$

Note that the equivariant linear weight blocks $\{W_b\}_{b=0}^{G+k}$ lie in the same subspace, which corresponds to the nullspace of the coefficient matrix $C$. We can obtain it via SVD.

We summarize the architecture of the EKAN layer in Figure 2 (Right). In this example, the EKAN layer is equivariant with respect to a 2-dimensional matrix group $\widetilde{G}$ (such as the SO(2) group). The user specifies the input space $U_i = T_0 \oplus T_1$ (where we abbreviate $T(p, 0) = V^p$ as $T_p$), which represents a scalar space and a vector space, and specifies the output space $U_o = T_2$, which represents a matrix space. Then, the gated input space $U_{gi} = T_0 \oplus T_1 \oplus T_0$ and the gated output space $U_{go} = T_2 \oplus T_0$ each add a gate scalar to the vector space $T_1$ and the matrix space $T_2$. The basis functions are applied to the gate scalars of $T_0$ (itself) and $T_1$, which are then multiplied by the original terms to obtain the post-activation space $U_m = 3U_i$. The linear weights $W \in \mathbb{R}^{5 \times 9}$ between $U_m$ and $U_{go}$ are within the subspace determined by Equation (20) to ensure equivariance. Similar to KANs (Liu et al., 2024b), EKAN updates grids based on the input activations, which we discuss in detail in Appendix E.

## 5. EKAN Architecture

In this section, we construct the entire EKAN architecture. The main body of EKAN is composed of stacked EKAN layers. The output space of the $l$-th layer serves as the input

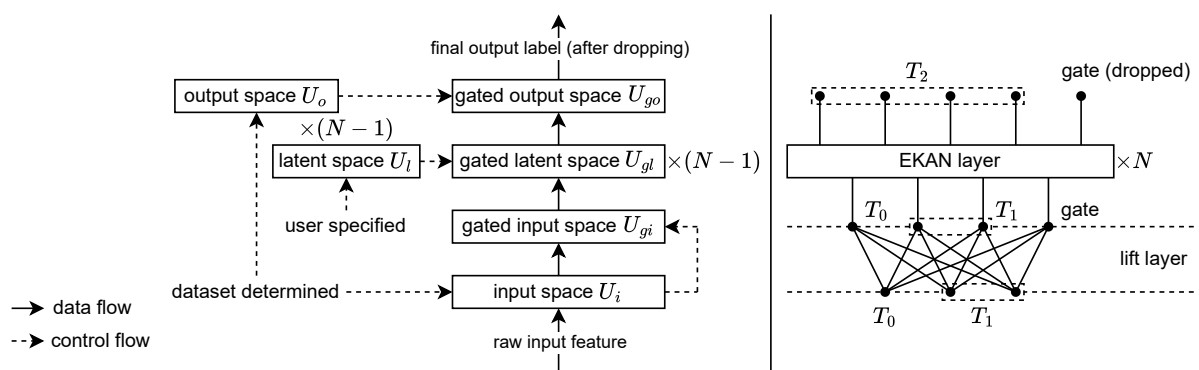

*Figure 3.* (Left) The space structures of EKAN and their relationships. (Right) The EKAN architecture, which consists of a lift layer and stacked EKAN layers.

space of the $(l + 1)$-th layer, which we refer to as the latent space $U_l$. For the dataset, we usually know its data type, or in other words, how group elements act on it. To embed this prior knowledge into EKAN, we set the input space of the first layer as the feature space of the dataset $U_i$ and the output space of the final layer as the label space of the dataset $U_o$.

However, the actual input/output features of the EKAN layer stack lie in $U_{gi}/U_{go}$. Therefore, we need to add extensions to align the network with the dataset. First, the gate scalars of the actual output feature are directly dropped to obtain the final output label of EKAN, which resides in $U_o$. Then, we add a lift layer before the first layer to preprocess the raw input feature of EKAN, which is essentially an equivariant linear layer between $U_i$ and $U_{gi}$ (see Section 4.3).

We summarize the space structures and network architecture of EKAN in Figure 3. The space structures of EKAN can be analogous to the dimensions of a conventional network. The user specifies the latent space $U_l$ of EKAN, which corresponds to specifying the hidden dimension in a conventional network. The input space $U_i$ and the output space $U_o$ are determined by the dataset, similar to how the input and output dimensions are defined in a conventional network. In the concrete example, the feature space and label space of the dataset are $U_i = T_0 \oplus T_1$ and $U_o = T_2$, respectively. After passing through the lift layer, a new gate scalar is added to the raw input feature for $T_1$, resulting in the actual input feature $U_{gi} = T_0 \oplus T_1 \oplus T_0$ for the first EKAN layer. The gate scalar in the actual output space $U_{go} = T_2 \oplus T_0$ of the last EKAN layer is dropped to obtain the final output label.

## 6. Experiments

In this section, we evaluate the performance of EKAN on regression and classification tasks with known symmetries. Compared with general models such as MLPs, KANs, and

equivariant architectures like EMLP (Finzi et al., 2021) and CGENN (Ruhe et al., 2023), EKAN achieves lower test loss and higher test accuracy with smaller datasets or fewer parameters. Additionally, for all trained models in this section, we evaluate their equivariant errors in Appendix F.

### 6.1. Particle Scattering

In electron-muon scattering, we can observe the four-momenta of the incoming electron, incoming muon, outgoing electron, and outgoing muon, denoted as $q^\mu, p^\mu, \tilde{q}^\mu, \tilde{p}^\mu \in \mathbb{R}^4$, respectively. We aim to predict the matrix element, which is proportional to the cross-section (Finzi et al., 2021): $|\mathcal{M}|^2 \propto [p^\mu \tilde{p}^\nu - (p^\alpha \tilde{p}_\alpha - p^\alpha p_\alpha)g^{\mu\nu}][q_\mu \tilde{q}_\nu - (q^\alpha \tilde{q}_\alpha - q^\alpha q_\alpha)g_{\mu\nu}]$. According to Einstein's summation convention, in a monomial, if an index appears once as a superscript and once as a subscript, it indicates summation over that index. The metric tensor is given by $g_{\mu\nu} = g^{\mu\nu} = \text{diag}(1, -1, -1, -1)$, and $a_\mu = g_{\mu\nu}a^\nu = (a^0, -a^1, -a^2, -a^3)$. The matrix element is invariant under Lorentz transformations. In other words, this task exhibits $O(1, 3)$ invariance (see Appendix A.2 for more details), with the feature space $U_i = 4T_1$ and the label space $U_o = T_0$. The data generation process for particle scattering is entirely consistent with that in EMLP (Finzi et al., 2021).

We embed the group $O(1, 3)$ and its subgroups $SO^+(1, 3)$ and $SO(1, 3)$ equivariance into EKAN. Models are evaluated on synthetic datasets with different training set sizes, which are generated by sampling $q^\mu, p^\mu, \tilde{q}^\mu, \tilde{p}^\mu \sim \mathcal{N}(0, \frac{1}{4^2})$. Both EKAN and KAN have the depth of $L = 2$, the spline order of $k = 3$, and grid intervals of $G = 3$. Although the lift layer increases the parameter overhead, we set the width of the middle layer in EKAN to $n_1 = 1000$ (shape as $[16, 1000, 1]$, and the software will automatically calculate the appropriate feature space structure based on the user-specified dimension), and set the width of the mid-

*Table 1.* Test MSE of different models on the particle scattering dataset with different training set sizes. We present the results in the format of mean ± std.

| Models | Training set size | | | | |
|---|---|---|---|---|---|
| | $10^2$ | $10^{2.5}$ | $10^3$ | $10^{3.5}$ | $10^4$ |
| MLP | $(7.33 \pm 0.01) \times 10^{-1}$ | $(6.97 \pm 0.09) \times 10^{-1}$ | $(3.64 \pm 0.30) \times 10^{-1}$ | $(5.04 \pm 0.37) \times 10^{-2}$ | $(1.66 \pm 0.07) \times 10^{-2}$ |
| MLP + augmentation | $(1.90 \pm 0.22) \times 10^{-1}$ | $(3.97 \pm 0.52) \times 10^{-2}$ | $(1.25 \pm 0.08) \times 10^{-2}$ | $(1.08 \pm 0.03) \times 10^{-2}$ | $(1.34 \pm 0.22) \times 10^{-2}$ |
| EMLP-SO$^+$(1, 3) | $(1.27 \pm 0.35) \times 10^{-2}$ | $(2.21 \pm 0.56) \times 10^{-3}$ | $(3.30 \pm 0.86) \times 10^{-4}$ | $(2.24 \pm 0.55) \times 10^{-4}$ | $(1.99 \pm 0.33) \times 10^{-4}$ |
| EMLP-SO(1, 3) | $(1.47 \pm 0.91) \times 10^{-2}$ | $(2.58 \pm 0.25) \times 10^{-3}$ | $(3.69 \pm 1.25) \times 10^{-4}$ | $(2.73 \pm 0.30) \times 10^{-4}$ | $(2.12 \pm 0.15) \times 10^{-4}$ |
| EMLP-O(1, 3) | $(8.88 \pm 2.51) \times 10^{-3}$ | $(1.95 \pm 0.18) \times 10^{-3}$ | $(3.30 \pm 0.43) \times 10^{-4}$ | $(2.66 \pm 0.66) \times 10^{-4}$ | $(2.64 \pm 0.28) \times 10^{-4}$ |
| KAN | $(6.70 \pm 1.35) \times 10^{-1}$ | $(6.16 \pm 1.18) \times 10^{-1}$ | $(3.46 \pm 0.15) \times 10^{-1}$ | $(1.21 \pm 0.07) \times 10^{-1}$ | $(2.57 \pm 0.08) \times 10^{-2}$ |
| KAN + augmentation | $(5.98 \pm 0.67) \times 10^{-1}$ | $(4.72 \pm 1.52) \times 10^{-1}$ | $(9.07 \pm 2.36) \times 10^{-2}$ | $(5.61 \pm 0.70) \times 10^{-2}$ | $(1.58 \pm 0.06) \times 10^{-1}$ |
| EKAN-SO$^+$(1, 3) (Ours) | $\mathbf{(6.86 \pm 6.28) \times 10^{-3}}$ | $(1.85 \pm 1.75) \times 10^{-3}$ | $\mathbf{(2.01 \pm 1.93) \times 10^{-5}}$ | $(1.93 \pm 1.11) \times 10^{-5}$ | $(4.29 \pm 3.38) \times 10^{-6}$ |
| EKAN-SO(1, 3) (Ours) | $\mathbf{(6.86 \pm 6.27) \times 10^{-3}}$ | $(1.85 \pm 1.75) \times 10^{-3}$ | $(2.06 \pm 1.88) \times 10^{-5}$ | $(2.17 \pm 1.51) \times 10^{-5}$ | $(3.85 \pm 2.77) \times 10^{-6}$ |
| EKAN-O(1, 3) (Ours) | $(7.77 \pm 5.85) \times 10^{-3}$ | $\mathbf{(1.64 \pm 1.87) \times 10^{-3}}$ | $(2.85 \pm 3.09) \times 10^{-5}$ | $\mathbf{(7.31 \pm 4.15) \times 10^{-6}}$ | $\mathbf{(3.81 \pm 2.83) \times 10^{-6}}$ |

*Table 2.* Test MSE of different models with different numbers of parameters on the three-body problem dataset. We present the results in the format of mean ± std.

| Models | Number of parameters | | | | |
|---|---|---|---|---|---|
| | $10^{4.5}$ | $10^{4.75}$ | $10^5$ | $10^{5.25}$ | $10^{5.5}$ |
| MLP | $(4.84 \pm 0.19) \times 10^{-3}$ | $(4.70 \pm 0.30) \times 10^{-3}$ | $(4.60 \pm 0.12) \times 10^{-3}$ | $(4.17 \pm 0.24) \times 10^{-3}$ | $(4.24 \pm 0.27) \times 10^{-3}$ |
| MLP + augmentation | $(9.43 \pm 0.88) \times 10^{-3}$ | $(9.65 \pm 0.54) \times 10^{-3}$ | $(1.01 \pm 0.09) \times 10^{-2}$ | $(9.89 \pm 0.65) \times 10^{-3}$ | $(9.91 \pm 0.46) \times 10^{-3}$ |
| EMLP-SO(2) | $(2.28 \pm 1.17) \times 10^{-3}$ | $(6.87 \pm 5.29) \times 10^{-3}$ | $(3.55 \pm 1.59) \times 10^{-3}$ | $(2.01 \pm 1.09) \times 10^{-3}$ | $(5.34 \pm 3.78) \times 10^{-3}$ |
| EMLP-O(2) | $(7.72 \pm 8.71) \times 10^{-3}$ | $(1.18 \pm 0.22) \times 10^{-3}$ | $(1.42 \pm 1.86) \times 10^{-2}$ | $(7.37 \pm 7.60) \times 10^{-3}$ | $(1.37 \pm 0.07) \times 10^{-3}$ |
| KAN | $(4.32 \pm 3.08) \times 10^{-1}$ | $(2.21 \pm 0.65) \times 10^{-2}$ | $(1.18 \pm 0.18) \times 10^{-2}$ | $(1.23 \pm 0.34) \times 10^{-2}$ | $(9.15 \pm 1.76) \times 10^{-3}$ |
| KAN + augmentation | $(7.94 \pm 0.15) \times 10^{-3}$ | $(7.15 \pm 0.24) \times 10^{-3}$ | $(6.91 \pm 0.34) \times 10^{-3}$ | $(6.88 \pm 0.06) \times 10^{-3}$ | $(6.76 \pm 0.12) \times 10^{-3}$ |
| EKAN-SO(2) (Ours) | $\mathbf{(1.12 \pm 0.13) \times 10^{-3}}$ | $\mathbf{(7.06 \pm 0.65) \times 10^{-4}}$ | $\mathbf{(6.09 \pm 0.27) \times 10^{-4}}$ | $\mathbf{(4.26 \pm 0.19) \times 10^{-4}}$ | $\mathbf{(4.84 \pm 0.68) \times 10^{-4}}$ |
| EKAN-O(2) (Ours) | $(1.48 \pm 0.37) \times 10^{-3}$ | $(1.12 \pm 0.24) \times 10^{-3}$ | $(7.91 \pm 0.52) \times 10^{-4}$ | $(6.06 \pm 0.36) \times 10^{-4}$ | $(6.02 \pm 0.88) \times 10^{-4}$ |

dle layer in KAN to $n_1 = 3840$ (shape as $[16, 3840, 1]$) to keep the parameter count similar. Both EMLP and MLP have the depth of $L = 4$ and the middle layer width of $n_1 = n_2 = n_3 = 384$ (shape as $[16, 384, 384, 384, 1]$). In these settings, EKAN (435k) has fewer parameters than EMLP (450k) and KAN (461k). We provide more implementation details in Appendix G.1.

We repeat experiments with three different random seeds and report the mean ± std of the test MSE in Table 1. The results of EMLP and MLP come from the original paper (Finzi et al., 2021) under the same settings. Although EMLP performs better than non-equivariant models, our EKAN with different group equivariance further surpasses it comprehensively, especially showing an orders-of-magnitude advantage on large datasets (training set size $\geq 10^3$). Moreover, our EKAN with just $10^3$ training samples achieves approximately $10\%$ of the test MSE of EMLP with $10^4$ training samples.

## 6.2. Three-Body Problem

The study of the three-body problem on a plane (Greydanus et al., 2019) focuses on the motion of three particles, with their center of mass at the origin, under the influence of gravity. Their trajectories are chaotic and cannot be described by an analytical solution. Specifically, we observe the motion states of three particles over the past four time steps, denoted as $\{q_{i1}, p_{i1}, q_{i2}, p_{i2}, q_{i3}, p_{i3}\}_{i=t-4}^{t-1}$, and predict their motion states at time $t$, denoted as $\{q_{t1}, p_{t1}, q_{t2}, p_{t2}, q_{t3}, p_{t3}\}$. Here, $q_{ij} \in \mathbb{R}^2$ and $p_{ij} \in \mathbb{R}^2$ indicate the position and momentum coordinates of the $j$-th particle at time $i$, respectively. The dataset contains 30k training samples and 30k test samples. When the input motion states are simultaneously rotated by a certain angle or reflected along a specific axis, the output motion states should undergo the same transformation. Therefore, this task has O(2) equivariance (see Appendix A.1 for more details), with the feature space $U_i = 4 \times 6T_1 = 24T_1$ and the label space $U_o = 6T_1$. The dataset for the three-body problem comes from HNN (Greydanus et al., 2019), and we note that they predict the motion trajectories of three particles, differing from the setup in CGENN (Ruhe et al., 2023), which addresses an $N$-body problem involving five particles.

We embed the group O(2) and its subgroup SO(2) equivariance into EKAN and EMLP. Both EKAN and KAN have the depth of $L = 2$, the spline order of $k = 3$, and grid intervals of $G = 3$, while both EMLP and MLP have the depth of $L = 4$. The number of parameters is controlled by adjusting the middle layer width $N$ for comparison (the shape of EKAN and KAN is $[48, N, 12]$, while the shape of

*Table 3.* Test accuracy (%) of different models on the top quark tagging dataset ($n_{comp} = 3$) with different training set sizes. We present the results in the format of mean ± std.

| Models | Training set size | | | | | Parameters |
|---|---|---|---|---|---|---|
| | $10^2$ | $10^{2.5}$ | $10^3$ | $10^{3.5}$ | $10^4$ | |
| MLP | $52.96 \pm 0.21$ | $54.31 \pm 0.48$ | $57.47 \pm 0.32$ | $62.72 \pm 0.60$ | $69.30 \pm 1.03$ | 83K |
| MLP + augmentation | $52.72 \pm 0.35$ | $60.16 \pm 0.41$ | $61.29 \pm 1.66$ | $58.71 \pm 1.37$ | $59.45 \pm 1.90$ | |
| EMLP-SO$^+(1,3)$ | $65.48 \pm 1.21$ | $72.59 \pm 0.84$ | $74.40 \pm 0.26$ | $76.34 \pm 0.14$ | $77.10 \pm 0.02$ | |
| EMLP-SO$(1,3)$ | $61.86 \pm 5.92$ | $73.09 \pm 0.92$ | $74.37 \pm 0.17$ | $76.46 \pm 0.12$ | $\mathbf{77.12 \pm 0.04}$ | 133K |
| EMLP-O$(1,3)$ | $62.66 \pm 7.35$ | $73.65 \pm 1.01$ | $74.22 \pm 0.53$ | $76.26 \pm 0.05$ | $\mathbf{77.12 \pm 0.04}$ | |
| KAN | $49.89 \pm 0.39$ | $49.91 \pm 0.43$ | $49.89 \pm 0.37$ | $50.00 \pm 0.02$ | $49.84 \pm 0.25$ | 35K |
| KAN + augmentation | $49.83 \pm 0.28$ | $50.09 \pm 0.09$ | $49.73 \pm 0.40$ | $50.02 \pm 0.01$ | $50.27 \pm 0.67$ | |
| CGENN | $62.63 \pm 2.24$ | $68.74 \pm 0.77$ | $70.29 \pm 1.29$ | $75.10 \pm 0.47$ | $77.05 \pm 0.03$ | 85K |
| EKAN-SO$^+(1,3)$ (Ours) | $\mathbf{71.92 \pm 0.88}$ | $\mathbf{73.98 \pm 0.39}$ | $\mathbf{76.15 \pm 0.11}$ | $\mathbf{76.69 \pm 0.08}$ | $76.93 \pm 0.02$ | |
| EKAN-SO$(1,3)$ (Ours) | $70.49 \pm 2.85$ | $73.96 \pm 0.37$ | $\mathbf{76.15 \pm 0.11}$ | $\mathbf{76.69 \pm 0.08}$ | $76.93 \pm 0.02$ | 34K |
| EKAN-O$(1,3)$ (Ours) | $71.68 \pm 1.21$ | $73.95 \pm 0.36$ | $\mathbf{76.15 \pm 0.11}$ | $\mathbf{76.69 \pm 0.07}$ | $76.93 \pm 0.03$ | |

EMLP and MLP is $[48, N, N, N, 12]$). More implementation details can be found in Appendix G.2.

The mean ± std of the test MSE over three runs with different random seeds are reported in Table 2. Our EKAN-SO$(2)$ and EKAN-O$(2)$ consistently outperform baseline models with the same number of parameters, often by orders of magnitude. Notably, our EKAN with $10^{4.5}$ parameters achieves comparable or even lower test MSE than baseline models with $10^{5.5}$ parameters, saving 90% of the parameter overhead.

### 6.3. Top Quark Tagging

The research on top quark tagging (Kasieczka et al., 2019) involves classifying hadronic tops from the QCD background. In particle collision experiments, top quark decays or other events produce several jet constituents. We observe the four-momenta $p_1^\mu, p_2^\mu, p_3^\mu \in \mathbb{R}^4$ of the three jet constituents with the highest transverse momentum $p_T$, and predict the event label (1 for top, 0 for QCD). The category of the event will not change when all jet constituents undergo the same Lorentz transformation. Consequently, this task possesses O$(1,3)$ invariance (see Appendix A.2 for more details), with the feature space $U_i = 3T_1$ and the label space $U_o = T_0$. The top quark tagging dataset is sourced from Kasieczka et al. (2019), and we assume that only the $n_{comp} = 3$ jet constituents with the highest transverse momentum $p_T$ are observed, which differs from the setup in CGENN (Ruhe et al., 2023) and LorentzNet (Gong et al., 2022), where all $n_{comp} = 200$ jet constituents are available.

Similar to particle scattering, we embed the group O$(1,3)$ and its subgroups SO$^+(1,3)$ and SO$(1,3)$ equivariance into EKAN and EMLP. Furthermore, we embed O$(1,3)$ equivariance into CGENN (Ruhe et al., 2023). We sample training

sets of different sizes from the original dataset for evaluation. Both EKAN and KAN have the depth of $L = 2$, the spline order of $k = 3$, and grid intervals of $G = 3$. We set the width of the middle layer in EKAN to $n_1 = 200$ (shape as $[12, 200, 1]$) and the width of the middle layer in KAN to $n_1 = 384$ (shape as $[12, 384, 1]$) to control the number of parameters. Both MLP, EMLP and CGENN have the depth of $L = 4$ and the middle layer width $n_1 = n_2 = n_3 = 200$ (shape as $[12, 200, 200, 200, 1]$). We apply the sigmoid function to the model's output and use BCE as the loss function for binary classification. More implementation details are provided in Appendix G.3.

We report the mean ± std of the test accuracy over three runs with different random seeds, as well as the number of parameters of the models in Table 3. Since we have not observed all the jet constituents, the relationship between the labels and input features cannot be accurately expressed as an explicit function. In this case of non-symbolic formula representation, KAN cannot achieve higher accuracy with fewer parameters than MLP as expected. On the other hand, our EKAN achieves comparable results with EMLP and CGENN using fewer than 40% of the parameters, improving test accuracy by $0.23\% \sim 6.44\%$ on small datasets (training set size $< 10^4$), while decreasing by $0.19\%$ on large datasets (training set size $= 10^4$).

## 7. Conclusion

To our knowledge, this work is the first attempt to combine equivariance and KANs. We view the KAN layer as a combination of spline functions and linear weights, and accordingly define the (gated) input space, post-activation space, and (gated) output space of the EKAN layer. Gated basis functions ensure the equivariance between the gated

input space and the post-activation space, while equivariant linear weights guarantee the equivariance between the post-activation space and the gated output space. The prior work has demonstrated that "EMLP > MLP" on tasks with symmetries and "KAN > MLP" on symbolic formula representation tasks. Our experimental results further indicate that on symbolic formula representation tasks with symmetries, "EKAN > EMLP" and "EKAN > KAN". Moreover, on non-symbolic formula representation tasks with symmetries, although it may be that "KAN < MLP", we show that "EKAN > EMLP". We expect that EKAN can become a general framework for applying KANs to more fields, such as computer vision and natural language processing, just as EMLP unifies classic works like CNNs and DeepSets.

## 8. Limitations

One limitation of our approach is that the use of gating mechanisms can reduce the expressive power of EKAN, as discussed in Appendix D of EMLP (Finzi et al., 2021). However, due to the inherently complex structure of KANs (which involve intricate B-spline formulations), introducing symmetries into KANs is challenging. Therefore, for the sake of simplicity and clarity, we opt to incorporate gated non-linearities in a hierarchical manner. We anticipate that future improvements could enhance the expressive power and flexibility of EKAN.

Another limitation is that when data is sufficient, the advantages of EKAN diminish compared with baselines. We provide additional experimental results in Appendix H to illustrate this point. This implies that EKAN enhances generalization on small data but does not unlock the upper limit of expressive power.

## Acknowledgements

Z. Lin was supported by National Key R&D Program of China (2022ZD0160300) and the NSF China (No. 62276004).

## Impact Statement

This paper presents work whose goal is to advance the field of Machine Learning. There are many potential societal consequences of our work, none which we feel must be specifically highlighted here.

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

## A. Common Matrix Groups and Their Generators

### A.1. Groups $\mathrm{SO}(2)$ and $\mathrm{O}(2)$

The group $\mathrm{SO}(2)$ represents rotation transformations in two-dimensional space. Its group elements can be expressed as:

$$R(\theta) = \begin{bmatrix} \cos\theta & -\sin\theta \\ \sin\theta & \cos\theta \end{bmatrix}. \tag{21}$$

It corresponds to an infinitesimal generator:

$$A_1 = \begin{bmatrix} 0 & -1 \\ 1 & 0 \end{bmatrix}. \tag{22}$$

Then we can obtain the group elements through the exponential map $R(\theta) = \exp(\theta A_1)$.

The group $\mathrm{O}(2)$ represents orthogonal transformations in two-dimensional space, including rotations and reflections. Based on the group $\mathrm{SO}(2)$, it has an additional discrete generator:

$$h_1 = \begin{bmatrix} 1 & 0 \\ 0 & -1 \end{bmatrix}. \tag{23}$$

### A.2. Groups $\mathrm{SO}^+(1,3)$, $\mathrm{SO}(1,3)$, and $\mathrm{O}(1,3)$

The group $\mathrm{SO}^+(1,3)$ represents Lorentz transformations that preserve both orientation and the direction of time. It includes six infinitesimal generators:

$$\left\{ \begin{aligned} A_1 &= \begin{bmatrix} 0 & 1 & 0 & 0 \\ 1 & 0 & 0 & 0 \\ 0 & 0 & 0 & 0 \\ 0 & 0 & 0 & 0 \end{bmatrix}, A_2 = \begin{bmatrix} 0 & 0 & 1 & 0 \\ 0 & 0 & 0 & 0 \\ 1 & 0 & 0 & 0 \\ 0 & 0 & 0 & 0 \end{bmatrix}, A_3 = \begin{bmatrix} 0 & 0 & 0 & 1 \\ 0 & 0 & 0 & 0 \\ 0 & 0 & 0 & 0 \\ 1 & 0 & 0 & 0 \end{bmatrix}, \\ A_4 &= \begin{bmatrix} 0 & 0 & 0 & 0 \\ 0 & 0 & 1 & 0 \\ 0 & -1 & 0 & 0 \\ 0 & 0 & 0 & 0 \end{bmatrix}, A_5 = \begin{bmatrix} 0 & 0 & 0 & 0 \\ 0 & 0 & 0 & 1 \\ 0 & 0 & 0 & 0 \\ 0 & -1 & 0 & 0 \end{bmatrix}, A_6 = \begin{bmatrix} 0 & 0 & 0 & 0 \\ 0 & 0 & 0 & 0 \\ 0 & 0 & 0 & 1 \\ 0 & 0 & -1 & 0 \end{bmatrix}, \end{aligned} \right. \tag{24}$$

where $A_1, A_2, A_3$ correspond to Lorentz boosts, and $A_4, A_5, A_6$ correspond to spatial rotations.

The group $\mathrm{SO}(1,3)$ represents Lorentz transformations that preserve orientation. Based on the group $\mathrm{SO}^+(1,3)$, it has an additional discrete generator:

$$h_1 = \begin{bmatrix} -1 & 0 & 0 & 0 \\ 0 & -1 & 0 & 0 \\ 0 & 0 & -1 & 0 \\ 0 & 0 & 0 & -1 \end{bmatrix}, \tag{25}$$

which corresponds to orientation reversal.

The group $\mathrm{O}(1,3)$ represents all Lorentz transformations. Based on the group $\mathrm{SO}(1,3)$, it has an additional discrete generator:

$$h_2 = \begin{bmatrix} -1 & 0 & 0 & 0 \\ 0 & 1 & 0 & 0 \\ 0 & 0 & 1 & 0 \\ 0 & 0 & 0 & 1 \end{bmatrix}, \tag{26}$$

which corresponds to time reversal.

## B. Concrete Examples of Space Structure

First, let's intuitively understand the dual ($*$), direct sum ($\oplus$), and tensor product ($\otimes$) operations. Consider two vector spaces $X = \mathbb{R}^2, Y = \mathbb{R}^3$, and vectors $x = (x_1, x_2) \in X, y = (y_1, y_2, y_3) \in Y$. Then, all $x \oplus y = (x_1, x_2, y_1, y_2, y_3)$ form the

space $X \oplus Y = \mathbb{R}^5$, all $x \otimes y = (x_1 y, x_2 y) = (x_1 y_1, x_1 y_2, x_1 y_3, x_2 y_1, x_2 y_2, x_2 y_3)$ form the space $X \otimes Y = \mathbb{R}^6$, and all the coefficients $\text{vec}(W)$ of the linear maps $Wx = y$ form the space $Y \otimes X^* = \mathbb{R}^6$.

Then, if we define how the group transformation acts on $X, Y$, the form of its action on these composite spaces can naturally be derived. Equation (4) provides the derivation rules. This paper assumes the group to be a matrix group. If a group element $g \in G$ acts on a vector $x \in X$ in the form of its corresponding linear transformation $gx$, then we call $X$ the base vector space of $G$.

We have defined the "addition" and "multiplication" between spaces. Thus, for the base vector space $V$ of a group $G$, any complex space structure can be organized into the form of a "polynomial" with respect to $V$, which is the origin of Equation (3). Note that Equation (3) simultaneously defines $T(p, q) = V^p \otimes (V^*)^q$. We abbreviate $T(p, 0)$ as $T_p$.

In the vast majority of scenarios, the feature space of a dataset takes simple forms such as vector stacking $cT_1 = cV$ or matrices $T_2 = V \otimes V$, while complex spaces like $T(p, q)$ rarely appear. However, the latent spaces between equivariant layers can be highly intricate (e.g., they may have $384$ dimensions), and their decomposition forms with respect to the base vector space $V$ often involve $T(p, q)$ (the decomposition is automatically handled by the software based on dimensionality, as described in Section 6). The practical implication is that, according to the rules of Equation (4), we define how group transformations operate on the latent spaces.

It is not correct that any (continuous, real, finite-dimensional) representation $U$ of a matrix group can be written as in Equation (3). It can be shown (in the case of a compact Lie group for example) that $U$ is a subrepresentation of the direct sum on the right hand side. However, it is worth noting that, for the vast majority of practical applications, considering input/output representations of this form should suffice.

## C. Equivariant Multi-Layer Perceptrons (EMLP)

EMLP (Finzi et al., 2021) embeds matrix group equivariance into MLPs layerwise. Given the input space $U_i$ and output space $U_o$, the linear weight matrix $W \in U_o \otimes U_i^*$ should satisfy Equation (5), i.e., $\forall g \in \widetilde{G}, v_i \in U_i : \rho_o(g)Wv_i = W\rho_i(g)v_i$. So the coefficients of each term in $v_i$ are equal $\forall g \in \widetilde{G} : \rho_o(g)W = W\rho_i(g)$. Flattening the linear weight matrix $W$ into a vector, we have $\forall g \in \widetilde{G} : [\rho_o(g) \otimes \rho_i(g^{-1})^\top] \text{vec}(W) = \text{vec}(W)$. Combined with Equation (4), the linear weight matrix $W$ is invariant in the space $U_o \otimes U_i^*$:

$$\forall g \in \widetilde{G} : \quad \rho_{o,i}(g)\text{vec}(W) = \text{vec}(W), \tag{27}$$

where $\rho_{o,i} = \rho_o \otimes \rho_i^*$ is the group representation of $U_o \otimes U_i^*$. Decomposing the group representation $\rho_{o,i}(g)$ into discrete and infinitesimal generators as shown in Equation (2), Equation (27) is equivalent to the following constraint:

$$C\text{vec}(W) = \begin{bmatrix} \mathrm{d}\rho_{o,i}(A_1) \\ \vdots \\ \mathrm{d}\rho_{o,i}(A_D) \\ \rho_{o,i}(h_1) - I \\ \vdots \\ \rho_{o,i}(h_M) - I \end{bmatrix} \text{vec}(W) = 0. \tag{28}$$

By performing singular value decomposition (SVD) on the coefficient matrix $C$, we can obtain its nullspace, which corresponds to the subspace where the equivariant linear weights reside.

## D. Proof of Theorem 4.1

**Theorem 4.1.** *Given a matrix group $\widetilde{G}$, the gated input space $U_{gi}$ and the post-activation space $U_m$ can be expressed in the forms of Equations (9) and (13), respectively. The function $f : U_{gi} \to U_m$ is defined by Equations (10) to (12), that is, $v_m = f(v_{gi})$. Then, $f$ is equivariant:*

$$\forall g \in \widetilde{G}, v_{gi} \in U_{gi} : \quad \rho_m(g)f(v_{gi}) = f(\rho_{gi}(g)v_{gi}), \tag{14}$$

*where $\rho_{gi}$ and $\rho_m$ are group representations of $U_{gi}$ and $U_m$, respectively.*

*Proof.* Let $v_{m,b} = f_b(v_{gi})$, then Equation (11) can be written as:

$$f(v_{gi}) = \bigoplus_{b=0}^{G+k} f_b(v_{gi}). \tag{29}$$

Using the structure of $U_m$ shown in Equation (13) and applying the rules from Equation (4), we can derive the group representation of $U_m$:

$$\rho_m(g) = \bigoplus_{b=0}^{G+k} \rho_i(g), \tag{30}$$

where $\rho_i$ is the group representation of $U_i$. Combining Equations (29) and (30), we have:

$$\rho_m(g)f(v_{gi}) = \bigoplus_{b=0}^{G+k} \rho_i(g)f_b(v_{gi}). \tag{31}$$

Note that the group transformation in the scalar space $T_0$ is the identity transformation, then we can obtain the group representation of $U_{gi}$ from Equation (9):

$$\rho_{gi}(g) = I_{c_i} \oplus \left[ \bigoplus_{a=1}^{A_i} \rho_{i,a}(g) \right] \oplus I_{A_i}, \tag{32}$$

where $\rho_{i,a}$ is the group representation of $T(p_{i,a}, q_{i,a})$. Therefore, applying the group transformation to $v_{gi}$ in Equation (10) results in:

$$\rho_{gi}(g)v_{gi} = \left( \bigoplus_{a=1}^{c_i} s_{i,a} \right) \oplus \left( \bigoplus_{a=1}^{A_i} \rho_{i,a}(g)v_{i,a} \right) \oplus \left( \bigoplus_{a=1}^{A_i} s'_{i,a} \right). \tag{33}$$

Substitute Equation (33) into Equation (12):

$$f_b(\rho_{gi}(g)v_{gi}) = \begin{cases} [\bigoplus_{a=1}^{c_i} s_{i,a}B_b(s_{i,a})] \oplus \left[\bigoplus_{a=1}^{A_i} \rho_{i,a}(g)v_{i,a}B_b(s'_{i,a})\right], & b < G+k, \\ [\bigoplus_{a=1}^{c_i} s_{i,a}\mathrm{silu}(s_{i,a})] \oplus \left[\bigoplus_{a=1}^{A_i} \rho_{i,a}(g)v_{i,a}\mathrm{silu}(s'_{i,a})\right], & b = G+k. \end{cases} \tag{34}$$

Similar to Equation (32), we can derive the group representation of $U_i$ from Equation (8):

$$\rho_i(g) = I_{c_i} \oplus \left[ \bigoplus_{a=1}^{A_i} \rho_{i,a}(g) \right]. \tag{35}$$

Note that the right-hand side of Equation (34) is the result of applying $\rho_i(g)$ to $f_b(v_{gi})$, which means:

$$f_b(\rho_{gi}(g)v_{gi}) = \rho_i(g)f_b(v_{gi}). \tag{36}$$

Substitute Equation (36) into Equation (29):

$$f(\rho_{gi}(g)v_{gi}) = \bigoplus_{b=0}^{G+k} \rho_i(g)f_b(v_{gi}). \tag{37}$$

Combining Equations (31) and (37), Equation (14) is proven. $\qquad\square$

## E. Grid Update

Similar to KANs (Liu et al., 2024b), EKAN updates grids based on the input activations. At the same time, the linear weights should also be updated in order to keep the output features unchanged. Let the post-activation values of the grids before and after the update be denoted as $V_m, V'_m \in \mathbb{R}^{N \times (G+k+1)d_i}$, where $N$ is the number of samples. We first project the linear weight blocks into the equivariant subspace as $\mathrm{vec}(\widetilde{W}_b) = P\mathrm{vec}(W_b)$ and compute the output activations $V_{go} = V_m \widetilde{W}^\top = \sum_{b=0}^{G+k} V_{m,b}\widetilde{W}_b^\top$, where $P$ is the equivariant projector obtained from Equation (20). Then, the updated equivariant linear weights $\widetilde{W}'$ should satisfy $V_{go} = V'_m \widetilde{W}'^\top$, and we have $\widetilde{W}' = V_{go}^\top (V_m'^\top)^\dagger$. We finally restore the updated linear weight blocks $\mathrm{vec}(W'_b) = P^\dagger \mathrm{vec}(\widetilde{W}'_b)$.

# F. Equivariant Error Evaluation

For all trained models $f_\theta$ in Section 6, we use $\mathcal{L}_{equi} = \mathbb{E}_{x,g} \|\rho_o(g)f_\theta(x) - f_\theta(\rho_i(g)x)\|^2$ to evaluate their equivariant errors. The experimental results are presented in Tables 4 to 6, which indicate that our EKAN and EMLP can structurally guarantee strict equivariance, whereas non-equivariant models cannot. For particle scattering, the results of EMLP and MLP are sourced from the original paper (Finzi et al., 2021), so we do not present their equivariant errors here.

*Table 4.* Equivariant error of different models on the particle scattering dataset with different training set sizes. We present the results in the format of mean $\pm$ std.

| Models | Training set size | | | | |
| | $10^2$ | $10^{2.5}$ | $10^3$ | $10^{3.5}$ | $10^4$ |
|---|---|---|---|---|---|
| KAN | $(3.14 \pm 0.08) \times 10^{-1}$ | $1.02 \pm 0.69$ | $(4.88 \pm 2.02) \times 10^{-1}$ | $(6.90 \pm 3.25) \times 10^{-1}$ | $(1.64 \pm 0.41) \times 10^{-1}$ |
| KAN + augmentation | $(2.78 \pm 0.30) \times 10^{-1}$ | $(8.81 \pm 8.41) \times 10^{-1}$ | $(1.08 \pm 0.55) \times 10^{-1}$ | $(1.32 \pm 0.52) \times 10^{-1}$ | $(2.36 \pm 0.17) \times 10^{-1}$ |
| EKAN-SO$^+$(1,3) (Ours) | $(6.13 \pm 1.76) \times 10^{-14}$ | $(9.32 \pm 2.75) \times 10^{-14}$ | $(7.48 \pm 1.61) \times 10^{-14}$ | $(8.33 \pm 0.69) \times 10^{-14}$ | $(6.57 \pm 0.06) \times 10^{-14}$ |
| EKAN-SO(1,3) (Ours) | $(1.13 \pm 0.78) \times 10^{-13}$ | $(7.64 \pm 1.67) \times 10^{-14}$ | $(6.86 \pm 0.11) \times 10^{-14}$ | $(7.98 \pm 0.45) \times 10^{-14}$ | $(6.62 \pm 0.89) \times 10^{-14}$ |
| EKAN-O(1,3) (Ours) | $(5.67 \pm 0.87) \times 10^{-14}$ | $(7.95 \pm 2.67) \times 10^{-14}$ | $(5.88 \pm 0.81) \times 10^{-14}$ | $(9.00 \pm 0.63) \times 10^{-14}$ | $(8.43 \pm 0.18) \times 10^{-14}$ |

*Table 5.* Equivariant error of different models with different numbers of parameters on the three-body problem dataset. We present the results in the format of mean $\pm$ std.

| Models | Number of parameters | | | | |
| | $10^{4.5}$ | $10^{4.75}$ | $10^5$ | $10^{5.25}$ | $10^{5.5}$ |
|---|---|---|---|---|---|
| MLP | $(1.44 \pm 0.13) \times 10^{-3}$ | $(1.51 \pm 0.14) \times 10^{-3}$ | $(1.54 \pm 0.03) \times 10^{-3}$ | $(1.33 \pm 0.17) \times 10^{-3}$ | $(1.33 \pm 0.10) \times 10^{-3}$ |
| MLP + augmentation | $(3.16 \pm 0.32) \times 10^{-3}$ | $(3.47 \pm 0.24) \times 10^{-3}$ | $(3.43 \pm 0.24) \times 10^{-3}$ | $(3.35 \pm 0.16) \times 10^{-3}$ | $(3.32 \pm 0.26) \times 10^{-3}$ |
| EMLP-SO(2) | $(2.80 \pm 2.18) \times 10^{-13}$ | $(1.57 \pm 1.39) \times 10^{-13}$ | $(2.17 \pm 0.37) \times 10^{-14}$ | $(2.99 \pm 2.51) \times 10^{-14}$ | $(1.87 \pm 0.61) \times 10^{-14}$ |
| EMLP-O(2) | $(7.87 \pm 6.26) \times 10^{-13}$ | $(1.87 \pm 1.32) \times 10^{-12}$ | $(3.22 \pm 4.53) \times 10^{-11}$ | $(1.46 \pm 2.03) \times 10^{-12}$ | $(7.23 \pm 6.13) \times 10^{-14}$ |
| KAN | $(3.00 \pm 2.28) \times 10^{-1}$ | $(1.21 \pm 0.26) \times 10^{-2}$ | $(5.90 \pm 0.78) \times 10^{-3}$ | $(5.93 \pm 1.45) \times 10^{-3}$ | $(4.03 \pm 0.81) \times 10^{-3}$ |
| KAN + augmentation | $(2.78 \pm 0.10) \times 10^{-3}$ | $(2.49 \pm 0.11) \times 10^{-3}$ | $(2.37 \pm 0.11) \times 10^{-3}$ | $(2.35 \pm 0.11) \times 10^{-3}$ | $(2.29 \pm 0.08) \times 10^{-3}$ |
| EKAN-SO(2) (Ours) | $(3.80 \pm 3.84) \times 10^{-13}$ | $(3.02 \pm 3.05) \times 10^{-13}$ | $(9.66 \pm 5.85) \times 10^{-14}$ | $(3.33 \pm 1.47) \times 10^{-14}$ | $(3.31 \pm 1.31) \times 10^{-14}$ |
| EKAN-O(2) (Ours) | $(9.79 \pm 6.13) \times 10^{-13}$ | $(1.46 \pm 1.30) \times 10^{-13}$ | $(2.99 \pm 2.77) \times 10^{-13}$ | $(1.75 \pm 1.55) \times 10^{-13}$ | $(7.62 \pm 4.91) \times 10^{-14}$ |

*Table 6.* Equivariant error of different models on the top quark tagging dataset ($n_{comp} = 3$) with different training set sizes. We present the results in the format of mean $\pm$ std.

| Models | Training set size | | | | |
| | $10^2$ | $10^{2.5}$ | $10^3$ | $10^{3.5}$ | $10^4$ |
|---|---|---|---|---|---|
| MLP | $(4.94 \pm 0.40) \times 10^{-1}$ | $(4.00 \pm 0.17) \times 10^{-1}$ | $(3.73 \pm 0.05) \times 10^{-1}$ | $(3.24 \pm 0.11) \times 10^{-1}$ | $(1.87 \pm 0.03) \times 10^{-1}$ |
| MLP + augmentation | $(4.73 \pm 0.28) \times 10^{-1}$ | $(2.15 \pm 0.12) \times 10^{-1}$ | $(1.90 \pm 0.55) \times 10^{-1}$ | $(3.73 \pm 0.39) \times 10^{-1}$ | $(3.40 \pm 0.63) \times 10^{-1}$ |
| EMLP-SO$^+$(1,3) | $(2.92 \pm 2.30) \times 10^{-6}$ | $(1.09 \pm 0.84) \times 10^{-7}$ | $(3.88 \pm 1.44) \times 10^{-9}$ | $(3.54 \pm 2.47) \times 10^{-9}$ | $(1.65 \pm 2.11) \times 10^{-8}$ |
| EMLP-SO(1,3) | $(6.81 \pm 8.20) \times 10^{-7}$ | $(1.36 \pm 1.82) \times 10^{-7}$ | $(5.49 \pm 2.92) \times 10^{-9}$ | $(2.69 \pm 1.48) \times 10^{-9}$ | $(1.71 \pm 0.36) \times 10^{-9}$ |
| EMLP-O(1,3) | $(2.41 \pm 1.68) \times 10^{-7}$ | $(1.86 \pm 1.80) \times 10^{-7}$ | $(6.20 \pm 0.59) \times 10^{-9}$ | $(1.14 \pm 1.36) \times 10^{-8}$ | $(1.26 \pm 0.28) \times 10^{-9}$ |
| KAN | $(1.36 \pm 1.78) \times 10^{-1}$ | $(1.35 \pm 1.77) \times 10^{-1}$ | $(1.34 \pm 1.77) \times 10^{-1}$ | $(9.90 \pm 14.00) \times 10^{-5}$ | $(1.27 \pm 1.79) \times 10^{-1}$ |
| KAN + augmentation | $(1.20 \pm 1.61) \times 10^{-1}$ | $(2.10 \pm 2.97) \times 10^{-3}$ | $(1.10 \pm 1.51) \times 10^{-1}$ | $(1.52 \pm 2.15) \times 10^{-4}$ | $(1.44 \pm 1.78) \times 10^{-1}$ |
| EKAN-SO$^+$(1,3) (Ours) | $(3.41 \pm 2.99) \times 10^{-7}$ | $(1.66 \pm 0.85) \times 10^{-8}$ | $(1.62 \pm 0.51) \times 10^{-9}$ | $(1.65 \pm 1.01) \times 10^{-9}$ | $(1.64 \pm 1.53) \times 10^{-9}$ |
| EKAN-SO(1,3) (Ours) | $(3.10 \pm 1.81) \times 10^{-7}$ | $(1.09 \pm 0.60) \times 10^{-8}$ | $(1.11 \pm 0.41) \times 10^{-9}$ | $(1.13 \pm 0.70) \times 10^{-9}$ | $(1.14 \pm 1.10) \times 10^{-9}$ |
| EKAN-O(1,3) (Ours) | $(3.14 \pm 2.32) \times 10^{-7}$ | $(1.46 \pm 0.79) \times 10^{-8}$ | $(1.50 \pm 0.46) \times 10^{-9}$ | $(1.50 \pm 0.92) \times 10^{-9}$ | $(1.54 \pm 1.50) \times 10^{-9}$ |

# G. Implementation Details

## G.1. Particle Scattering

In particle scattering, we generate training sets of different sizes, and the corresponding test sets have the same sizes as the training sets. We train EKAN using the Adan optimizer (Xie et al., 2024) with the learning rate of $3 \times 10^{-3}$ and the batch size of 500. For datasets with the training set size $< 1000$, we set the number of epochs to 7000, while for datasets with the training set size $\geq 1000$, we set the number of epochs to 15000, which is sufficient for the MSE loss to converge to the

minimum. We perform this experiment on a single-core NVIDIA GeForce RTX 3090 GPU with available memory of 24576 MiB.

### G.2. Three-Body Problem

In the three-body problem, we control the number of parameters by adjusting the middle layer width $N$ of the model. We list the correspondence between the model's shape and the number of parameters in Table 7. We train all models using the Adan optimizer (Xie et al., 2024) with the learning rate of $3 \times 10^{-3}$, the batch size of 500, and for 5000 epochs. The grids of EKAN and KAN are updated every 5 epochs and stop updating at the 50th epoch. We perform this experiment on a single-core NVIDIA GeForce RTX 3090 GPU with available memory of 24576 MiB.

*Table 7.* The correspondence between the model's shape and the number of parameters.

| Models | Number of parameters | | | | |
| | $10^{4.5}$ | $10^{4.75}$ | $10^5$ | $10^{5.25}$ | $10^{5.5}$ |
|---|---|---|---|---|---|
| MLP | $[48, 111, 111, 111, 12]$ | $[48, 153, 153, 153, 12]$ | $[48, 209, 209, 209, 12]$ | $[48, 283, 283, 283, 12]$ | $[48, 383, 383, 383, 12]$ |
| EMLP | $[48, 84, 84, 84, 12]$ | $[48, 110, 110, 110, 12]$ | $[48, 147, 147, 147, 12]$ | $[48, 214, 214, 214, 12]$ | $[48, 281, 281, 281, 12]$ |
| KAN | $[48, 76, 12]$ | $[48, 134, 12]$ | $[48, 238, 12]$ | $[48, 423, 12]$ | $[48, 752, 12]$ |
| EKAN (Ours) | $[48, 45, 12]$ | $[48, 88, 12]$ | $[48, 151, 12]$ | $[48, 262, 12]$ | $[48, 457, 12]$ |

### G.3. Top Quark Tagging

In top quark tagging, we train all models using the Adan optimizer (Xie et al., 2024) with the learning rate of $3 \times 10^{-3}$ and the batch size of 500. For datasets with the training set size $\leq 1000$, we set the number of epochs to 1000, while for datasets with the training set size $> 1000$, we set the number of epochs to 2000, which is sufficient for the BCE loss to converge to the minimum. The grids of EKAN and KAN are updated every 5 epochs and stop updating at the 50th epoch. We perform this experiment on a single-core NVIDIA GeForce RTX 3090 GPU with available memory of 24576 MiB.

## H. Additional Experiments

For top quark tagging, we increase the number of observed jet components for further comparison. We present the results using the $SO^+(1, 3)$-equivariant network as a representative, while the results of $SO(1, 3)$ and $O(1, 3)$-equivariant networks are very similar. We set the number of jet components to $n_{comp} = 10$ and $n_{comp} = 20$ respectively, and the experimental results are shown in Tables 8 and 9. Together with Table 3 in Section 6, we note that all models exhibit significant improvements in accuracy as the observed information increases. When $n_{comp}$ is larger, our EKAN achieves comparable results to the baselines, but it does not show superior performance. This suggests that EKAN may only have advantages in scenarios with insufficient observation information (as shown in Table 3), benefiting from its stronger generalization capability.

*Table 8.* Test accuracy (%) of different models on the top quark tagging dataset ($n_{comp} = 10$) with different training set sizes. We present the results in the format of mean $\pm$ std.

| Models | Training set size | | | | |
| | $10^2$ | $10^{2.5}$ | $10^3$ | $10^{3.5}$ | $10^4$ |
|---|---|---|---|---|---|
| EMLP-$SO^+(1, 3)$ | $\mathbf{78.95 \pm 2.24}$ | $\mathbf{81.52 \pm 0.48}$ | $81.18 \pm 0.42$ | $82.50 \pm 0.30$ | $85.03 \pm 0.04$ |
| CGENN | $71.82 \pm 3.28$ | $80.35 \pm 0.57$ | $79.85 \pm 1.01$ | $81.56 \pm 0.23$ | $84.17 \pm 0.76$ |
| EKAN-$SO^+(1, 3)$ (Ours) | $78.57 \pm 0.63$ | $79.77 \pm 0.78$ | $\mathbf{82.90 \pm 0.61}$ | $\mathbf{84.83 \pm 0.19}$ | $\mathbf{87.14 \pm 0.03}$ |

*Table 9.* Test accuracy (%) of different models on the top quark tagging dataset ($n_{comp} = 20$) with different training set sizes. We present the results in the format of mean $\pm$ std.

| Models | Training set size | | | | |
|---|---|---|---|---|---|
| | $10^2$ | $10^{2.5}$ | $10^3$ | $10^{3.5}$ | $10^4$ |
| EMLP-SO$^+$(1, 3) | $\mathbf{83.71 \pm 0.49}$ | $\mathbf{83.57 \pm 0.69}$ | $\mathbf{83.14 \pm 0.35}$ | $85.00 \pm 0.35$ | $86.81 \pm 0.14$ |
| CGENN | $76.24 \pm 1.28$ | $82.34 \pm 0.80$ | $81.85 \pm 0.40$ | $84.67 \pm 0.87$ | $86.76 \pm 0.50$ |
| EKAN-SO$^+$(1, 3) (Ours) | $80.36 \pm 1.99$ | $81.25 \pm 0.70$ | $82.89 \pm 1.18$ | $\mathbf{86.21 \pm 0.21}$ | $\mathbf{89.30 \pm 0.12}$ |

