# OpenReview forum: "Incorporating Arbitrary Matrix Group Equivariance into KANs"
_ICML.cc/2025/Conference — ICML 2025 poster_

### Official Review · Reviewer_yJMe · 2025-03-04

**Overall Recommendation:** 3

**Summary:**

This paper introduces Equivariant Kolmogorov-Arnold Networks (EKANs), an extension of Kolmogorov-Arnold Networks (KANs) that incorporates matrix group equivariance. The authors follow a similar approach as Equivariant MLPs (EMLP) by Marc Finzi et al. (2021) to enforce equivariance constraints on KANs. Basically, the weight is enforced to be in the intertwiner of the given group representations through a set of linear constraints. In addition, a lift layer is introduced to preprocess the input features, aligning them with the equivariant structure, while a gating mechanism ensures equivariance in the non-linear activation stage. The authors evaluate EKANs on tasks involving known symmetries (e.g., particle scattering, three-body problem, and top quark tagging), showing that EKANs achieve lower error and require fewer parameters compared to baseline models such as KANs, EMLPs, and other equivariant architectures.

**Claims And Evidence:**

The main claim of the paper is that EKANs improve upon both KANs and other equivariant architectures by leveraging equivariance constraints in a more efficient manner. The empirical results support the claim that EKANs outperform standard KANs and EMLPs in terms of test error reduction and parameter efficiency.

However, the key methodological innovation—incorporating equivariance via equivariant linear layers—is a direct adaptation of the approach used in EMLP (Finzi et al., 2021). The primary difference is that this method is now applied to KANs rather than MLPs. The authors claim novelty by applying equivariance to spline-based architectures, but they do not introduce a fundamentally new method for enforcing equivariance. This aspect is not sufficiently acknowledged in the paper.

**Essential References Not Discussed:**

N/A

**Experimental Designs Or Analyses:**

The experimental results appear sound, and the comparisons are generally fair. EKANs are compared to multiple baselines, including MLPs, EMLPs, standard KANs, and other relevant equivariant architectures. Also, the study varies the dataset size and the model capacity to analyze their influence on performance.

However, the paper does not explore settings where equivariance may not be useful or could degrade performance. It is noted in the EMLP paper that equivariant linear layers and gated nonlinearities are not universal, and there exist simple equivariant functions for some groups that cannot be approximated by EMLP. Since EKAN directly adopts the EMLP technique, it is likely that these limitations still exist and should be discussed in the paper.

Also, for the top tagging experiment, the authors have not included the comparison with GNN-based methods, e.g. LorentzNet (Gong et al., 2022),  which generally have higher accuracies.

### References
* Gong, Shiqi, et al. "An efficient Lorentz equivariant graph neural network for jet tagging." Journal of High Energy Physics 2022.7 (2022): 1-22.

**Methods And Evaluation Criteria:**

The proposed modifications to KANs follow a well-established framework for imposing equivariance. The evaluation primarily focuses on:
* Comparing EKANs against standard KANs, MLPs, and EMLPs on symmetry-sensitive tasks.
* Measuring test error across different dataset sizes and numbers of parameters.

The benchmarks are chosen from EMLP and other related works. The chosen benchmarks are reasonable, and the paper provides sufficient details on experimental settings.

**Other Comments Or Suggestions:**

N/A

**Other Strengths And Weaknesses:**

N/A

**Questions For Authors:**

N/A

## Post-rebuttal

One of my previous concerns is the novelty of this paper, compared to existing works such as EMLP. After careful reconsideration, I think incorporating equivariance into KAN is a valid and substantial contribution by itself, though the paper uses similar approaches to EMLP. I have updated my score to reflect this.

It should still be noted that the additional experiment results show that the advantage of EKAN diminishes with more data. EKAN may only be a better choice when data is relatively scarce.

**Relation To Broader Scientific Literature:**

This work heavily builds upon prior studies in equivariant networks and KANs. The foundational ideas stem from:
* EMLPs (Finzi et al., 2021), which already established a general approach for incorporating equivariance into MLPs. EKANs directly adopt this approach for KANs without substantial modification.
* KANs (Liu et al., 2024), which introduced spline-based learnable activation functions. The paper correctly identifies the limitations of KANs in handling symmetry constraints but does not present a fundamentally new way of addressing them.

Overall, the paper positions itself as an application of existing equivariant principles to a new network architecture rather than a significant methodological advance. While the application to KANs may be useful, the extent of novelty is limited.

**Theoretical Claims:**

N/A

---

> ### Author Rebuttal · Authors · 2025-04-01
>
> Thank you for your careful reading and valuable feedback! Below we will address each of your concerns point by point.
>
> **Claims And Evidence**
>
> > However, the key methodological innovation—incorporating equivariance via equivariant linear layers—is a direct adaptation of the approach used in EMLP (Finzi et al., 2021). The primary difference is that this method is now applied to KANs rather than MLPs. The authors claim novelty by applying equivariance to spline-based architectures, but they do not introduce a fundamentally new method for enforcing equivariance. This aspect is not sufficiently acknowledged in the paper.
>
> Due to the complexity of B-spline function formulations, embedding strict equivariance into KANs is challenging. Hierarchically employing gating mechanisms is a relatively straightforward approach, although we acknowledge this does have expressive limitations. Our claim is that we aim to introduce equivariance into KANs to address their poor performance on symmetric tasks and non-symbolic representation tasks, while preserving their advantages over MLPs. Thus, as an improvement to KANs, we strive to retain their original hierarchical structure and B-spline basis function form. From another perspective, if major structural modifications were made to propose an entirely new architecture, how would it remain related to KANs? Additionally, EKAN is not merely a simple fusion of KANs and traditional equivariant mechanisms—the design of the EKAN layer's post-activation space structure is also challenging (see Sections 4.1 and 4.2 for details).
>
> **Experimental Designs Or Analyses**
>
> > However, the paper does not explore settings where equivariance may not be useful or could degrade performance. It is noted in the EMLP paper that equivariant linear layers and gated nonlinearities are not universal, and there exist simple equivariant functions for some groups that cannot be approximated by EMLP. Since EKAN directly adopts the EMLP technique, it is likely that these limitations still exist and should be discussed in the paper.
>
> Indeed, the use of gating mechanisms can reduce the expressive power of the network. However, due to the inherently complex structure of KANs (which involve intricate B-spline formulations), introducing symmetry into KANs is challenging. Therefore, for the sake of simplicity and clarity, we opted to incorporate gated non-linearities in a hierarchical manner. We anticipate that future improvements could enhance the expressive power and flexibility of EKAN. We will include a discussion of this limitation and potential future work in the paper. Thank you for your suggestion!
>
> > Also, for the top tagging experiment, the authors have not included the comparison with GNN-based methods, e.g. LorentzNet (Gong et al., 2022), which generally have higher accuracies.
>
> Thank you for your addition! LorentzNet [1] achieves good results in top quark tagging, but it specifically focuses on this particular task. EKAN aims to provide a general framework for all symmetric scientific problems, so methods targeted at downstream tasks are not our primary benchmark for comparison. For example, CGENN [2] can incorporate more symmetries beyond the Lorentz group and outperforms LorentzNet when observations are complete (with 200 components). Therefore, in our paper, we chose to compare with CGENN. As we claimed, our EKAN outperforms CGENN in both accuracy and parameter efficiency when observations are incomplete (with only 3 components and a smaller dataset). Combining EKAN with task-specific models remains a direction for future research.
>
> **Supplementary Material**
>
> > There is no instruction on how to run the experiments, and I did not review the code in detail.
>
> The instructions for running the experiments are provided in the README.md document within the supplementary materials zip file. We will also make the code publicly available upon acceptance.
>
> **Relation To Broader Scientific Literature**
>
> > Overall, the paper positions itself as an application of existing equivariant principles to a new network architecture rather than a significant methodological advance. While the application to KANs may be useful, the extent of novelty is limited.
>
> Refer to the "Claims And Evidence" section in the Rebuttal.
>
> **Reference**
>
> [1] Gong, Shiqi, et al. "An efficient Lorentz equivariant graph neural network for jet tagging." Journal of High Energy Physics 2022.7 (2022): 1-22.
>
> [2] Ruhe, David, Johannes Brandstetter, and Patrick Forré. "Clifford group equivariant neural networks." Advances in Neural Information Processing Systems 36 (2023): 62922-62990.

---

> > ### Comment · Reviewer_yJMe · 2025-04-03
> >
> > Thank you for the response. Regarding the novelty, from the perspective of equivariant networks, this paper just follows an existing method to enforce equivariance. But I agree that incorporating symmetry into KAN is a valid contribution. I will reconsider this and possibly update my score.
> >
> > I still have questions about the top tagging experiment. You only used the three leading jet constituents, while other works (e.g. LorentzNet and CGENN) showed that using more constituents led to much higher accuracy. So what happens if you use more constituents? Does EKAN still have superior performance?
> >
> > I also suggest the authors highlight the differences in experimental setups and datasets from previous works. For example, I was unaware of the different number of jet constituents used in the top tagging experiment and was confused as to why the figures read differently from other works. Other reviewers have also mentioned something related to the n-body dataset, where you should mention the different dataset generation procedures.

---

> > > ### Author Response · Authors · 2025-04-04
> > >
> > > Thank you for your further reply and recognition of the contribution! Now we will try our best to address your remaining concerns.
> > >
> > > > I still have questions about the top tagging experiment. You only used the three leading jet constituents, while other works (e.g. LorentzNet and CGENN) showed that using more constituents led to much higher accuracy. So what happens if you use more constituents? Does EKAN still have superior performance?
> > >
> > > For top quark tagging, we increase the number of observed jet components for further comparison. We present the results using the $SO^+(1,3)$-equivariant network as a representative, while the results of $SO(1,3)$ and $O(1,3)$-equivariant networks are very similar. We set the number of jet components to $n_{comp}=10$ and $n_{comp}=20$ respectively, and the experimental results are shown below. Together with Table 3 in the paper, we observe that all models exhibit significant improvements in accuracy as the observed information increases. When $n_{comp}$ is larger, our EKAN achieves comparable results to the baselines. Indeed, EKAN does not show superior performance when the observation information is sufficient, but its advantage in scenarios with insufficient observation information (as shown in Table 3) highlights its stronger generalization capability.
> > >
> > > $n_{comp}=10$:
> > >
> > > |Models/Training set size|$10^2$|$10^{2.5}$|$10^3$|$10^{3.5}$|$10^4$|
> > > |-|-|-|-|-|-|
> > > |EMLP-$SO^+(1,3)$|$\mathbf{78.95\pm0.02}$|$\mathbf{81.52\pm0.48}$|$81.18\pm0.42$|$82.50\pm0.30$|$85.03\pm0.04$|
> > > |CGENN|$71.82\pm3.28$|$80.35\pm0.57$|$79.85\pm1.01$|$81.56\pm0.23$|$84.17\pm0.76$|
> > > |EKAN-$SO^+(1,3)$|$78.57\pm0.63$|$79.77\pm0.78$|$\mathbf{82.90\pm0.61}$|$\mathbf{84.83\pm0.20}$|$\mathbf{87.14\pm0.03}$|
> > >
> > > $n_{comp}=20$:
> > >
> > > |Models/Training set size|$10^2$|$10^{2.5}$|$10^3$|$10^{3.5}$|$10^4$|
> > > |-|-|-|-|-|-|
> > > |EMLP-$SO^+(1,3)$|$\mathbf{83.71\pm0.49}$|$\mathbf{83.57\pm0.69}$|$\mathbf{83.14\pm0.35}$|$85.00\pm0.35$|$86.81\pm0.14$|
> > > |CGENN|$76.24\pm1.28$|$82.34\pm0.80$|$81.85\pm0.40$|$84.67\pm0.87$|$86.76\pm0.50$|
> > > |EKAN-$SO^+(1,3)$|$80.36\pm1.99$|$81.25\pm0.70$|$82.89\pm1.18$|$\mathbf{86.21\pm0.21}$|$\mathbf{89.30\pm0.12}$|
> > >
> > > > I also suggest the authors highlight the differences in experimental setups and datasets from previous works. For example, I was unaware of the different number of jet constituents used in the top tagging experiment and was confused as to why the figures read differently from other works. Other reviewers have also mentioned something related to the n-body dataset, where you should mention the different dataset generation procedures.
> > >
> > > We will provide detailed explanations of the data sources, experimental setup, and the differences from the baseline's original paper in Section 6 (Experiments). Specifically, the data generation process for particle scattering is entirely consistent with that in EMLP [1]. The dataset for the three-body problem comes from HNN [2], and we note that they predict the motion trajectories of three particles, whereas CGENN [3] addresses a $N$-body problem involving five particles. The top quark tagging dataset is sourced from [4], and we assume that only the three jet constituents with the highest transverse momentum $p_T$ are observed (this has already been mentioned in Lines 409-412, column 1), which differs from [3, 5] where all 200 jet constituents are observed. Additionally, we will include the discussion related to the previous question (concerning $n_{comp}$) in the Appendix. Thank you for your suggestion!
> > >
> > > **Reference**
> > >
> > > [1] Finzi, Marc, Max Welling, and Andrew Gordon Wilson. "A practical method for constructing equivariant multilayer perceptrons for arbitrary matrix groups."
> > >
> > > [2] Greydanus, Samuel, Misko Dzamba, and Jason Yosinski. "Hamiltonian neural networks."
> > >
> > > [3] Ruhe, David, Johannes Brandstetter, and Patrick Forré. "Clifford group equivariant neural networks."
> > >
> > > [4] Kasieczka, Gregor, et al. "Top quark tagging reference dataset."
> > >
> > > [5] Gong, Shiqi, et al. "An efficient Lorentz equivariant graph neural network for jet tagging."

---

### Official Review · Reviewer_TCN4 · 2025-03-14

**Overall Recommendation:** 3

**Summary:**

This paper introduces Equivariant Kolmogorov-Arnold Networks, an extension of KANs that incorporates equivariance to arbitrary matrix groups, addressing a key limitation of KANs: their inability to respect symmetries in data. The authors achieve this by constructing gated spline basis functions and equivariant linear weights, ensuring the model remains equivariant throughout. A lift layer is introduced to preprocess inputs, aligning them with the dataset’s symmetry properties.

**Claims And Evidence:**

The motivation for the work is clear, and the theoretical claims are supported by the work. The extensive experiments also show an improvement over the nonequivaraint version. However, there is no validation as to whether the network is actually equivariant (either through an equivariant loss, or through the study of the representations), and the equivariance is only indirectly evaluated through performance.

**Essential References Not Discussed:**

There are no references (from the equivariance side) that I think are missing.

**Experimental Designs Or Analyses:**

The analyses are sensible and most observations seem insightful and accurate. Reiterating again my skepticism about the degree to which EKANs are structurally equivariant.

**Methods And Evaluation Criteria:**

The methods and evaluation criteria are sensible for the task. As mentioned above, the paper would be strengthened by showing EKANs are actually equivariant.

**Other Comments Or Suggestions:**

There are 3 figures that showcase the EKAN architecture, all are different, and actually none of them is explained in the text.

**Other Strengths And Weaknesses:**

The motivation is very clear and the work is interesting. Evaluation of the degree to which the model is actually equivariant is the main weakness. Also, the lifting layer could be discussed in greater depth: why is it necessary, why does it maintain equivariance, etc.

**Questions For Authors:**

No further questions, everything was addressed in the previous sections.

**Relation To Broader Scientific Literature:**

KANs have been very popular in the past year and equivariance has been steadily impactful over the last decade, so the work is of very broad interest.

**Theoretical Claims:**

I read the main theorem and it seemed reasonable, but did not thoroughly examine the proof. Similarly, I followed the derivations but did not thoroughly examine.

---

> ### Author Rebuttal · Authors · 2025-03-31
>
> Thank you for your careful reading and valuable feedback! Below we will address each of your concerns point by point.
>
> **Claims And Evidence**
>
> For all trained models $f_\theta$ in the paper, we use $L_{equi}=E_{x,g}\\|\rho_o(g)f_\theta(x)-f_\theta(\rho_i(g)x)\\|^2$ to evaluate their equivariant loss. The experimental results are presented as follows, which indicate that our EKAN and EMLP can structurally guarantee strict equivariance, whereas non-equivariant models cannot.
>
> Particle scattering (for this experiment, the results of EMLP and MLP are sourced from the original paper, so we do not present their equivariance errors here):
>
> |Models/Training set size|$10^2$|$10^{2.5}$|$10^3$|$10^{3.5}$|$10^4$|
> |-|-|-|-|-|-|
> |KAN|$(6.01\pm2.45)\times10^{-1}$|$1.11\pm0.63$|$(4.08\pm0.27)\times10^{-1}$|$(3.21\pm0.27)\times10^{-1}$|$(1.36\pm0.14)\times10^{-1}$|
> |KAN+augmentation|$(2.49\pm0.12)\times10^{-1}$|$(9.77\pm7.74)\times10^{-1}$|$(1.03\pm0.14)\times10^{-1}$|$(8.78\pm1.13)\times10^{-2}$|$(2.16\pm0.10)\times10^{-1}$|
> |EKAN|$(9.85\pm3.12)\times10^{-14}$|$(7.87\pm1.47)\times10^{-14}$|$(7.90\pm0.74)\times10^{-14}$|$(7.69\pm0.48)\times10^{-14}$|$(6.95\pm0.70)\times10^{-14}$|
>
> 3-body problem:
>
> |Models/Number of parameters|$10^{4.5}$|$10^{4.75}$|$10^5$|$10^{5.25}$|$10^{5.5}$|
> |-|-|-|-|-|-|
> |MLP|$(1.44\pm0.13)\times10^{-3}$|$(1.51\pm0.14)\times10^{-3}$|$(1.54\pm0.03)\times10^{-3}$|$(1.33\pm0.17)\times10^{-3}$|$(1.33\pm0.10)\times10^{-3}$|
> |MLP+aug|$(3.16\pm0.32)\times10^{-3}$|$(3.47\pm0.24)\times10^{-3}$|$(3.43\pm0.24)\times10^{-3}$|$(3.35\pm0.16)\times10^{-3}$|$(3.32\pm0.26)\times10^{-3}$|
> |EMLP|$(2.80\pm2.18)\times10^{-13}$|$(1.57\pm1.39)\times10^{-13}$|$(2.17\pm0.37)\times10^{-14}$|$(2.99\pm2.51)\times10^{-14}$|$(1.87\pm0.61)\times10^{-14}$|
> |KAN|$(3.00\pm2.28)\times10^{-1}$|$(1.21\pm0.26)\times10^{-2}$|$(5.90\pm0.78)\times10^{-3}$|$(5.93\pm1.45)\times10^{-3}$|$(4.03\pm0.81)\times10^{-3}$|
> |KAN+aug|$(2.78\pm0.10)\times10^{-3}$|$(2.49\pm0.11)\times10^{-3}$|$(2.37\pm0.11)\times10^{-3}$|$(2.35\times0.11)\times10^{-3}$|$(2.29\pm0.08)\times10^{-3}$|
> |EKAN|$(3.80\pm3.84)\times10^{-13}$|$(3.02\pm3.05)\times10^{-13}$|$(9.66\pm5.85)\times10^{-14}$|$(3.33\pm1.47)\times10^{-14}$|$(3.31\pm1.31)\times10^{-14}$|
>
> Top tagging:
>
> |Models/Training set size|$10^2$|$10^{2.5}$|$10^3$|$10^{3.5}$|$10^4$|
> |-|-|-|-|-|-|
> |MLP|$(4.94\pm0.40)\times10^{-1}$|$(4.00\pm0.17)\times10^{-1}$|$(3.73\pm0.05)\times10^{-1}$|$(3.24\pm0.11)\times10^{-1}$|$(1.87\pm0.03)\times10^{-1}$|
> |MLP+aug|$(4.73\pm0.28)\times10^{-1}$|$(2.15\pm0.12)\times10^{-1}$|$(1.90\pm0.55)\times10^{-1}$|$(3.73\pm0.39)\times10^{-1}$|$(3.40\pm0.63)\times10^{-1}$|
> |EMLP|$(2.92\pm2.30)\times10^{-6}$|$(1.09\pm0.84)\times10^{-7}$|$(3.88\pm1.44)\times10^{-9}$|$(3.54\pm2.47)\times10^{-9}$|$(1.64\pm1.53)\times10^{-9}$|
> |KAN|$(1.36\pm1.78)\times10^{-1}$|$(1.35\pm1.77)\times10^{-1}$|$(1.34\pm1.77)\times10^{-1}$|$(0.99\pm1.40)\times10^{-4}$|$(1.27\pm1.79)\times10^{-1}$|
> |KAN+aug|$(1.20\pm1.61)\times10^{-1}$|$(2.10\pm2.97)\times10^{-3}$|$(1.10\pm1.51)\times10^{-1}$|$(1.52\pm2.15)\times10^{-4}$|$(1.44\pm1.78)\times10^{-1}$|
> |EKAN|$(3.41\pm2.99)\times10^{-7}$|$(1.66\pm0.85)\times10^{-8}$|$(1.62\pm0.51)\times10^{-9}$|$(1.65\pm1.01)\times10^{-9}$|$(1.64\pm1.53)\times10^{-9}$|
>
> **Other Strengths And Weaknesses**
>
> As mentioned in the first paragraph of Section 5 (Lines 269-272, Column 2), EKAN is composed of stacked EKAN layers. The EKAN layer constructed in Section 4 has input space $U_{gi}$ and output space $U_{go}$ as shown in Equation (9). However, the feature space $U_i$ and label space $U_o$ of real-world datasets may not conform to this structure (specifically, they lack gate scalars). Therefore, as discussed in the second paragraph of Section 5 (Lines 295-299, Column 1), alignment is required. To align $U_{go}$ with $U_o$, we simply discard the gate scalars in $U_{go}$. For aligning $U_{gi}$ with $U_i$, we prepend a lift layer before the first EKAN layer to introduce gate scalars into $U_i$.
>
> The lift layer is essentially an equivariant linear layer between $U_i$ and $U_{gi}$ (as described in Lines 300-302, Column 1). Its specific construction is detailed in Section 4.3, ensuring that it inherently preserves equivariance. Additionally, in the Claims And Evidence section of the Rebuttal, we also experimentally validate that the entire EKAN architecture is strictly equivariant.
>
> **Other Comments Or Suggestions**
>
> Figure 1 provides a general comparison between the EKAN and KAN architectures, which we elaborate on in the third paragraph of Section 1 (Lines 45-54, Column 2, and Lines 55-66, Column 1). Figure 2 illustrates the structure of an individual EKAN layer, which we explain in detail in the last paragraph of Section 4.1 (Lines 214-219, Column 1, and Lines 180-182, Column 2). Figure 3 presents the overall architecture of EKAN, which we describe in the final paragraph of Section 5 (Lines 304-318, Column 1). We will make it clearer to avoid confusion—thank you for pointing this out!

---

### Official Review · Reviewer_nHH7 · 2025-03-14

**Overall Recommendation:** 3

**Summary:**

This paper introduces Equivariant Kolmogorov-Arnold Networks (EKANs), a framework to construct group equivariant architectures, with respect to arbitrary matrix groups, as an extension of the previously proposed Kolmogorov-Arnold networks, akin to the way Equivariant MLPs (EMLPs) extend conventional MLPs. Contrary to the linear blocks that are the heart of MLPs, KANs contain several components (spline basis functions, as well as silu activations) that are non-trivial to be converted to equivariant analogues. To address this, in a nutshell, the authors follow a gating mechanism strategy (similarly to EMLP) - input scalars are gate-transformed and then given as inputs to the basis functions and silu which are subsequently scalar multiplied to the input tensors. In that way input features are transformed to post-activations, in a manner that is proven to be equivariant. Finally, post-activations are linearly transformed to the output features, where the linear equivariant parameters are calculated using a method obtained from EMLP. Empirically, the method is shown to achieve improved results on various symmetry-related tasks,  even with fewer parameters and/or training samples, against previously proposed equivariant models, as well as standard KANs and MLPs.

**Claims And Evidence:**

The authors' central claim is that their architecture is an equivariant extension of KANs that improves upon baselines (e.g. MLP) with fewer parameters and fewer data. Equivariance is proven theoretically, while the rest of the claims are indeed convincingly supported by experiments.

**Essential References Not Discussed:**

Nothing to note.

**Experimental Designs Or Analyses:**

As discussed above, the experimental designs follow those of EMLP and their implementations and analyses seem sound.

**Methods And Evaluation Criteria:**

Since KANs have shown promise in other tasks, it is natural to extend them to symmetric problems. The benchmarks and evaluation criteria are obtained by EMLP, a known baseline in the literature. Therefore they are reasonable for the studied problem.

**Other Comments Or Suggestions:**

- To my knowledge,  it is not correct that any (continuous, real, finite-dimensional) representation $U$ of a matrix group can be written as in Equation (3). It can be shown  (in the case of a compact Lie group for example)  that  $U$  is a subrepresentation of the direct sum on the right hand side. However, it is worth noting that, for the vast majority of practical applications, considering input/output representations of this form should suffice.

- In lines 189-191, why are $p_{i,a}, p_{o,a}, q_{i,a}, q_{o,a}$ squared?

**Other Strengths And Weaknesses:**

**Strengths**

- The paper is well-structured and contains sufficient material on the background and related methods.
- The equivariant conversion of KANs is easy to implement and widely applicable.
- The empirical results demonstrate that the proposed method holds promise in real-world scenarios.

**Weaknesses**

- Apart from the improved experimental results provided in this paper, it is not evident how EKANs address the (i) scaling issues and (ii) limited expressivity (see Appendix D in Finzi et al., 2021) of Equivariant Multi-Layer Perceptrons (EMLPs) with gated non-linearities. Regarding (ii) it is unclear why the authors decided to resort to gating mechanisms for their construction and what are the implications of this choice.
- Additionally, in certain parts the method is hard to follow for the reader not versed in symmetries, in particular due to the fact that the notation is quite specialised and not quite intuitive. I believe that some concepts need to be simplified with some indicative examples. For example, Eq. 3 could be explained with a concrete example. Similarly, I am confused with the notation T(p, q), since in the experimental section only T(p,0) spaces are encountered. Can the authors give concrete examples here as well?

**Questions For Authors:**

- Besides the provided experimental results, are there  any advantages  of  EKANs  against  EMLPs  with  gated  non-linearities?  Could their scaling issues or limited expressivity, somehow be addressed?

- Does the linearity of the lift layer, between $U_i$ and $U_{gi}$, not prove problematic in some cases? For example, the only equivariant linear map from $T_1$ to $T_0$ is the constant zero, say, in the case of the orthogonal group (see Appendix D in Finzi  et al., 2021). This would mean that the gate scalars would always be zero in the gated input space.

- In the  experiments,  would  an  EMLP  with  fewer  parameters  achieve  potentially improved results? Having more parameters than the EKAN would probably mean that it is a more expressive model, prone to overfitting phenomena.

**Relation To Broader Scientific Literature:**

Constructing group equivariant neural architectures has enjoyed a fruitful line of research in recent years, and a wide range of applications. Notably, the first paragraph of the related work section of the paper includes some of the most prominent results in this direction. On
On the other hand, Kolmogorov-Arnold networks are a relatively new architecture, serving as a promising alternative to  Multi-Layer  Perceptrons.  The method proposed in this paper attempts to address the poor performance of KANs on certain tasks, owing to their difficulty in respecting the data type and symmetry,  as per the authors, by incorporating group equivariance in the KAN framework for the first time.

**Theoretical Claims:**

I studied the proof of Theorem 4.1, namely the equivariance of the proposed method, and I found it correct.

---

> ### Author Rebuttal · Authors · 2025-03-31
>
> Thank you for your careful reading and valuable feedback! Below we will address each of your concerns point by point.
>
> **Other Strengths And Weaknesses**
>
> Weaknesses
>
> (1) Indeed, the use of gating mechanisms can reduce the expressive power of the network. However, due to the inherently complex structure of KANs (which involve intricate B-spline formulations), introducing symmetry into KANs is challenging. Therefore, for the sake of simplicity and clarity, we opted to incorporate gated non-linearities in a hierarchical manner. We anticipate that future improvements could enhance the expressive power and flexibility of EKAN. We will include a discussion of this limitation and potential future work in the paper.
>
> (2) First, let's intuitively understand the dual ($\*$), direct sum ($\oplus$), and tensor product ($\otimes$) operations. Consider two vector spaces $X=R^2,Y=R^3$, and vectors $x=(x_1,x_2)\in X,y=(y_1,y_2,y_3)\in Y$. Then, all $x\oplus y=(x_1,x_2,y_1,y_2,y_3)$ form the space $X\oplus Y=R^5$, all $x\otimes y=(x_1y,x_2y)=(x_1y_1,x_1y_2,x_1y_3,x_2y_1,x_2y_2,x_2y_3)$ form the space $X\otimes Y=R^6$, and all the coefficients $vec(W)$ of the linear maps $Wx=y$ form the space $Y\otimes X^*=R^6$.
>
> Then, if we define how the group transformation acts on $X,Y$, the form of its action on these composite spaces can naturally be derived. Eqn (4) provides the derivation rule. The paper assumes the group to be a matrix group. If a group element $g\in G$ acts on a vector $x\in X$ in the form of its corresponding linear transformation $gx$, then we call $X$ the base vector space of $G$ (as described in Lines 102-105, Column 2).
>
> We have defined the "addition" and "multiplication" between spaces. Thus, for the base vector space $V$ of a group $G$, any complex spatial structure can be organized into the form of a "polynomial" with respect to $V$, which is the origin of Eqn (3). Note that Eqn (3) simultaneously defines $T(p,q)=V^p\otimes (V^*)^q$. We abbreviate $T(p,0)$ as $T_p$.
>
> In the vast majority of scenarios, the feature space of a dataset takes simple forms such as vector stacking $cT_1=cV$ or matrices $T_2=V\otimes V$, while complex spaces like $T(p,q)$ rarely appear. However, the latent spaces between equivariant layers can be highly intricate (e.g., they may have 384 dimensions), and their decomposition forms with respect to the base vector space $V$ often involve $T(p,q)$ (the decomposition is automatically handled by the software based on dimensionality, as described in Lines 317-320, Column 2). The practical implication is that, according to the rules of Eqn (4), we define how group transformations operate on the latent space.
>
> We will add these intuitive explanations to the appendix to avoid confusion. Thank you for your suggestion!
>
> **Other Comments Or Suggestions**
>
> (1) Thank you for pointing that out! We will note this in the paper to be more rigorous.
>
> (2) We aim to express that $p_{i,a}$ and $q_{i,a}$ are not simultaneously zero. Indeed, in the case where both are natural numbers, $p_{i,a}^2+q_{i,a}^2>0$ and $p_{i,a}+q_{i,a}>0$ are equivalent, with the latter being more concise. We will make the corresponding revision. Thank you!
>
> **Questions For Authors**
>
> (1) Refer to point (1) in the Other Strengths And Weaknesses section of the Rebuttal.
>
> (2) When the dimension of the latent space between equivariant layers is too small, its parameters may degenerate to all zeros. However, this issue rarely occurs when the latent space is more complex.
>
> (3) Overfitting often occurs with small datasets, so we reduced the number of parameters in EMLP and trained it on particle scattering with a relatively small training set size. The experimental results are shown below, which indicate that it does not perform as well as models with larger parameter sizes. In fact, such equivariant architectures are less prone to overfitting compared to non-equivariant models because they strictly respect the symmetries in the data.
>
> |Models/Training set size|$10^2$|$10^{2.5}$|
> |-|-|-|
> |EMLP-$SO^+(1,3)$ (210k parameters)|$(5.98\pm5.40)\times10^{-2}$|$(3.65\pm1.60)\times10^{-3}$|
> |EMLP-$SO(1,3)$ (210k parameters)|$(6.13\pm5.62)\times10^{-2}$|$(3.76\pm1.73)\times10^{-3}$|
> |EMLP-$O(1,3)$ (210k parameters)|$(6.14\pm5.71)\times10^{-2}$|$(3.64\pm1.64)\times10^{-3}$|
> |EMLP-$SO^+(1,3)$ (450k parameters)|$(1.27\pm0.35)\times10^{-2}$|$(2.21\pm0.56)\times10^{-3}$|
> |EMLP-$SO(1,3)$ (450k parameters)|$(1.47\pm0.91)\times10^{-2}$|$(2.58\pm0.25)\times10^{-3}$|
> |EMLP-$O(1,3)$ (450k parameters)|$(8.88\pm2.51)\times10^{-3}$|$(1.95\pm0.18)\times10^{-3}$|
> |EKAN-$SO^+(1,3)$ (435k parameters)|$\mathbf{(6.86\pm6.28)\times10^{-3}}$|$(1.85\pm1.75)\times10^{-3}$|
> |EKAN-$SO(1,3)$ (435k parameters)|$\mathbf{(6.86\pm6.27)\times10^{-3}}$|$(1.85\pm1.75)\times10^{-3}$|
> |EKAN-$O(1,3)$ (435k parameters)|$(7.77\pm5.85)\times10^{-3}$|$\mathbf{(1.64\pm1.87)\times10^{-3}}$|

---

### Official Review · Reviewer_oRYZ · 2025-03-14

**Overall Recommendation:** 2

**Summary:**

The work introduces an equivariant version of the KAN by incorporating two principal components: 1) introducing an additional scaler that controls the gating mechanism and 2) using equivariant MLP for different non-scaler features.

The proposed model, EKAN, is evaluated on particle scattering, top quark tagging, and three body problem datasets, and it outperformed EMLP in almost all the scenarios across different train dataset sizes.

**Claims And Evidence:**

The work is theoretically sound. The proposed architecture is equivariant with respect to the desired matrix group.

However, the claim regarding the superiority of the equivariant KAN compared to other models is ill-demonstrated.

This, in the end, depends on the following two probable interpretations of claims made.

Claim v1. **The work proposed an alternative to EMLP using KAN**: In this case, the experiments (with some additional details) support the claim. However, the scope becomes narrow.

Claim v2. **The work proposed a new Equivariant architecture:** In this case, more experiments are required. For example, E(3) GNN[1]  or [2] should be considered.

The work should state the claim more precisely.

[1]. Geometric and Physical Quantities Improve E(3) Equivariant Message Passing

[2]. Scalars are universal: Equivariant machine learning, structured like classical physics

**Essential References Not Discussed:**

N/A

**Experimental Designs Or Analyses:**

I have found the following issues and uninvestigated questions:
1. I do not find any details on the implementation of baselines.

2. Why the number of parameters of EMLP is very high? Does it follow the architecture proposed in the original paper? is the performance gap is due to overfitting? What would be the performance if both EKAN and EMLP had a similar number of parameters?

3. Why do we not consider equivariant GNN for three body problems? And it does not follow the exact setup of [a,b], i.e., five-body problem?




[a] Clifford Group Equivariant Neural Networks

[b] Geometric and Physical Quantities Improve E(3) Equivariant Message Passing

**Methods And Evaluation Criteria:**

The method and evaluation criteria are valid. Depending on the interpretation of the claim (if Claim v1), the choice of dataset is also reasonable.

**Other Comments Or Suggestions:**

N/A

**Other Strengths And Weaknesses:**

The work shows strong results compared to the EMLP baseline (some questions still need to be addressed).

Apart from limited novelty and evaluations, the writing of the paper can be significantly improved. For example, most of the topics discussed and notations introduced in the "Background" section are not used in Section 4 or in the main paper. For example, I do not believe the list in Eqn 4 is necessary to go through the main text. These can be moved to supplementary.

**Questions For Authors:**

No additional questions.

**Relation To Broader Scientific Literature:**

KAN is an emerging area in machine learning. This work introduces equivariance to the KAN framework, which broadens its applicability. However, from the perspective of equivariant neural networks, most of the techniques used in this work are known results.

**Theoretical Claims:**

The theoretical claim is correct, to my understanding.

---

> ### Author Rebuttal · Authors · 2025-04-01
>
> Thank you for your careful reading and valuable feedback! Below we will address each of your concerns point by point.
>
> **Claims And Evidence**
>
> Our claim is: We propose a method to incorporate symmetry into KANs. As mentioned in Section 1 (Lines 25-29, Column 2), KANs struggle to respect symmetry, which is one of the reasons for their underperformance in non-symbolic representation tasks. By introducing equivariant layers, we aim to address this limitation. Experimental results show that EKAN outperforms KANs on symmetry-related tasks while preserving KANs' advantages over MLPs, which supports our claim. Furthermore, even in scenarios where KANs are weaker than MLPs, EKAN still surpasses EMLP—an insight we expect to provide valuable contributions to this emerging topic of KANs.
>
> **Experimental Designs Or Analyses**
>
> (1) The model architecture of EMLP is exactly the same as in the original paper [1]. To control the number of parameters, we adjust its depth and width (i.e., shape), with relevant details provided in Section 6 of the main text (specifically the second paragraphs of Sections 6.1, 6.2, and 6.3). For fairness, EKAN and all baseline models follow identical training settings, which we elaborate on in Appendix E (Implementation Details). We will emphasize this in the paper to avoid confusion. Thank you for pointing it out!
>
> (2) Overfitting often occurs with small datasets, so we reduced the number of parameters in EMLP and trained it on particle scattering with a relatively small training set size. The experimental results are shown below, which indicate that it does not perform as well as models with larger parameter sizes. In fact, such equivariant architectures are less prone to overfitting compared to non-equivariant models because they strictly respect the symmetries in the data.
>
> |Models/Training set size|$10^2$|$10^{2.5}$|
> |-|-|-|
> |EMLP-$SO^+(1,3)$ (210k parameters)|$(5.98\pm5.40)\times10^{-2}$|$(3.65\pm1.60)\times10^{-3}$|
> |EMLP-$SO(1,3)$ (210k parameters)|$(6.13\pm5.62)\times10^{-2}$|$(3.76\pm1.73)\times10^{-3}$|
> |EMLP-$O(1,3)$ (210k parameters)|$(6.14\pm5.71)\times10^{-2}$|$(3.64\pm1.64)\times10^{-3}$|
> |EMLP-$SO^+(1,3)$ (450k parameters)|$(1.27\pm0.35)\times10^{-2}$|$(2.21\pm0.56)\times10^{-3}$|
> |EMLP-$SO(1,3)$ (450k parameters)|$(1.47\pm0.91)\times10^{-2}$|$(2.58\pm0.25)\times10^{-3}$|
> |EMLP-$O(1,3)$ (450k parameters)|$(8.88\pm2.51)\times10^{-3}$|$(1.95\pm0.18)\times10^{-3}$|
> |EKAN-$SO^+(1,3)$ (435k parameters)|$\mathbf{(6.86\pm6.28)\times10^{-3}}$|$(1.85\pm1.75)\times10^{-3}$|
> |EKAN-$SO(1,3)$ (435k parameters)|$\mathbf{(6.86\pm6.27)\times10^{-3}}$|$(1.85\pm1.75)\times10^{-3}$|
> |EKAN-$O(1,3)$ (435k parameters)|$(7.77\pm5.85)\times10^{-3}$|$\mathbf{(1.64\pm1.87)\times10^{-3}}$|
>
> (3) Our dataset for the three-body problem originates from LieGAN [2] rather than CGENN [3] (note that LieGAN is a method for symmetry discovery rather than the design of equivariant networks, so it is not our comparison target), so there are differences in dataset generation details and experimental setups. We supplement a comparison between CGENN and EKAN on the three-body problem, with the results shown below, which demonstrate EKAN's higher accuracy compared to CGENN.
>
> |Models/Number of parameters|$10^{4.5}$|$10^{4.75}$|$10^5$|$10^{5.25}$|$10^{5.5}$|
> |-|-|-|-|-|-|
> |CGENN|$(2.11\pm0.11)\times10^{-3}$|$(1.93\pm0.40)\times10^{-3}$|$(1.64\pm0.21)\times10^{-3}$|$(1.56\pm0.18)\times10^{-3}$|$(1.38\pm0.26)\times10^{-3}$|
> |EKAN-SO(2)|$\mathbf{(1.12\pm0.13)\times10^{-3}}$|$\mathbf{(7.06\pm0.65)\times10^{-4}}$|$\mathbf{(6.09\pm0.27)\times10^{-4}}$|$\mathbf{(4.26\pm0.19)\times10^{-4}}$|$\mathbf{(4.84\pm0.68)\times10^{-4}}$|
> |EKAN-O(2)|$(1.48\pm0.37)\times10^{-3}$|$(1.12\pm0.24)\times10^{-3}$|$(7.91\pm0.52)\times10^{-4}$|$(6.06\pm0.36)\times10^{-4}$|$(6.02\pm0.88)\times10^{-4}$|
>
> **Other Strengths And Weaknesses**
>
> The Background section primarily helps readers unfamiliar with symmetry theory to understand the relevant knowledge and avoid confusion. An important purpose of Eqn (4) is to inform readers that we decompose the space $U$ in the form of Eqn (3), whose practical significance is to define how group transformations act on $U$. This is crucial when constructing the latent space of the EKAN Layer (especially at the code implementation level), because the form of the group representation on the latent space depends on the definition provided by Eqn (4). More details can be found in point (2) of Other Strengths And Weaknesses section in the Rebuttal to Reviewer nHH7. In the revised version, we will move relatively less important concepts to the supplementary material. Thank you for your suggestion!
>
> **References**
>
> [1] Finzi et al. "A practical method for constructing equivariant multilayer perceptrons for arbitrary matrix groups."
>
> [2] Yang et al. "Generative adversarial symmetry discovery."
>
> [3] Ruhe et al. "Clifford group equivariant neural networks."

---

### Decision · Program_Chairs · 2025-05-01

**Decision:**

Accept (poster)

**Comment:**

This paper introduces Equivariant Kolmogorov-Arnold Networks (EKANs), extending KANs to incorporate arbitrary matrix group equivariance. After careful consideration of the reviews and author rebuttals, I recommend accepting this paper, although I note it is a borderline case. While concerns were raised about the novelty relative to existing equivariant architectures, the authors convincingly demonstrated that incorporating equivariance into the complex structure of KANs presents unique challenges that they've effectively addressed. The experimental results showing EKAN's superior performance with fewer parameters and training samples are compelling. We recommend the authors address the limitations related to expressivity and provide more detailed experimental comparisons in the final version.